# Voronoi-grid-based Pareto Front Learning and Its Application to Collaborative Federated Learning

Mengmeng Chen [* 1]  Xiaohu Wu [* 1]  Qiqi Liu [2]  Tiantian He [3]  Yew-Soon Ong [3 4]  Yaochu Jin [2]  Qicheng Lao [1]
Han Yu [4]

## Abstract

Multi-objective optimization (MOO) exists extensively in machine learning, and aims to find a set of Pareto-optimal solutions, called the Pareto front, e.g., it is fundamental for multiple avenues of research in federated learning (FL). Pareto-Front Learning (PFL) is a powerful method implemented using Hypernetworks (PHNs) to approximate the Pareto front. This method enables the acquisition of a mapping function from a given preference vector to the solutions on the Pareto front. However, most existing PFL approaches still face two challenges: (a) sampling rays in high-dimensional spaces; (b) failing to cover the entire Pareto Front which has a convex shape. Here, we introduce a novel PFL framework, called as PHN-HVVS, which decomposes the design space into Voronoi grids and deploys a genetic algorithm (GA) for Voronoi grid partitioning within high-dimensional space. We put forward a new loss function, which effectively contributes to more extensive coverage of the resultant Pareto front and maximizes the HV Indicator. Experimental results on multiple MOO machine learning tasks demonstrate that PHN-HVVS outperforms the baselines significantly in generating Pareto front. Also, we illustrate that PHN-HVVS advances the methodologies of several recent problems in the FL field. The code is available at https://github.com/buptcmm/phnhvvs.

---

*Equal contribution  [1]Beijing University of Posts and Telecommunications, China  [2]School of Engineering, Westlake University, China  [3]Centre for Frontier AI Research, Institute of High Performance Computing, Agency for Science, Technology and Research, Singapore  [4]College of Computing and Data Science, Nanyang Technological University, Singapore. Correspondence to: Han Yu <han.yu@ntu.edu.sg>, Xiaohu Wu <xiaohu.wu@bupt.edu.cn>.

*Proceedings of the 42$^{nd}$ International Conference on Machine Learning*, Vancouver, Canada. PMLR 267, 2025. Copyright 2025 by the author(s).

## 1. Introduction

Multi-objective optimization (MOO) is a critical area of research in machine learning, addressing problems that involve optimizing multiple conflicting objectives simultaneously. In real-world applications such as engineering design (Nedjah & Mourelle, 2015; Yi et al., 2021), financial planning (Nobre & Neves, 2019; Doumpos & Zopounidis, 2020), and resource allocation (Gong et al., 2019; Wu et al., 2019a; 2025; Ma et al., 2023), MOO problems are ubiquitous, where solutions need to balance trade-offs between competing goals. Pareto-Front Learning (PFL) has emerged as an effective technique to address these problems, utilizing Pareto HyperNetworks (PHNs) to generate solutions along the Pareto front. PFL allows the learning of a mapping function that takes a given preference vector as an input and outputs the corresponding solution on the Pareto front, providing a principled way to obtain optimal trade-offs between different objectives (Navon et al., 2020).

Despite the promising potential of PFL approaches, two significant limitations persist in existing methods: (a) difficulties in efficiently sampling rays in high-dimensional spaces, and (b) challenges in covering the entire Pareto front, which is essential for obtaining a comprehensive set of solutions. These limitations have been widely acknowledged in prior work. For instance, the difficulty of sampling in high-dimensional spaces has been highlighted in (Hoang et al., 2023). Similarly, the issue of inadequate Pareto front coverage has been extensively studied in (Hua et al., 2021; Zhang et al., 2023; Hoang et al., 2023). To address these issues, we propose a novel framework for PFL, termed PHN-HVVS. This framework novelly decomposes the design space into Voronoi grids (Aurenhammer & Klein, 2000), with the help of a genetic algorithm (GA) (Holland, 1992) for high-dimensional Voronoi grid partitioning. This partitioning technique allows for more effective exploration of the Pareto front.

Furthermore, we propose a new loss function within this framework, which significantly enhances the coverage of the Pareto front and maximizes the Hypervolume Indicator (HV) (Zitzler et al., 2007). HV metric can simultaneously evaluate the convergence and diversity of a set of solutions,

which refer to how closely the solutions approximate the true Pareto front (PF), and how well the solutions are spread across the entire PF, respectively. Through extensive experiments on various MOO machine learning tasks, we demonstrate that PHN-HVVS outperforms existing baselines by generating more well-balanced and diverse Pareto fronts, especially in convex problem scenarios. Our approach not only improves the coverage of the Pareto front but also enhances the computation of benefit graphs (Cui et al., 2022; Tan et al., 2024; Chen et al., 2024; Li et al., 2025), delivering notable improvements compared to traditional methods. This work lays the foundation for more effective and scalable MOO solutions in machine learning.

**Organization**. Section 2 provides an overview of the related works. In Section 3, we introduce the related techniques such as Voronoi Diagram and Pareto front learning for MOO. In Section 4, we describe the proposed PHN-HVVS framework in detail. In Section 5, we experimentally validate the effectiveness of PHN-HVVS. Finally, we conclude this paper in Section 6.

## 2. Related Work

**Applications of Voronoi Diagram in Multi-Objective Optimization** The Voronoi Diagram is a pivotal tool in MOO, facilitating the partition of spaces to enhance solution distributions and convergence. Its applications span various domains, including estimation of distribution algorithms (EDAs), airspace sector redesign, and zoning design. The Voronoi-based EDA (VEDA) (Okabe et al., 2004) leverages Voronoi cells to model solution distributions. It applies PCA to project high-dimensional spaces data into a low-dimensional space and then uses Voronoi grids. This approach outperforms traditional methods such as NSGA-II (Deb et al., 2002) in terms of adaptability and search efficiency. Combined with genetic algorithms (GAs), Voronoi Diagram partitions airspace sectors in 2D space to minimize multi-objective costs (Xue, 2009), achieving balanced airspace management. Similarly, in zoning design of 2D space, Pirlo & Impedovo (2012) utilizes Voronoi Diagram alongside multi-objective GA to determine optimal zone numbers and boundaries. These instances underscore the versatility and efficacy of Voronoi Diagram in addressing complex optimization challenges across diverse fields. However, applying Voronoi grids directly to high-dimensional MOO remains underexplored.

**Pareto front learning for Multi-Objective Optimization** In (Navon et al., 2020), the author proposes the new term Pareto-Front Learning (PFL) and describes an approach to PFL implemented using HyperNetworks which is termed as Pareto HyperNetworks (PHNs). PHNs is a more efficient and practical way since it learns the entire Pareto front simultaneously using a single hypernetwork. PHN-LS and PHN-EPO cover the Pareto front by varying the input preference vector. PHN-LS uses linear scalarization with the preference vector as the loss function. PHN-EPO uses the EPO (Mahapatra & Rajan, 2020) update direction for each output of PHNs. Co-PMTL (Lin et al., 2020) uses Pareto MTL to connect the preference vector and corresponding Pareto solution. HV (Zitzler et al., 2007), an important indicator for comparing the quality of different solution sets, is also utilized to optimize the hypernetwork. SEPNET (Chang et al., 2021) expresses PFL as a new MOO problem and then defines the fitness function as HV. HVmax (Deist et al., 2021) uses a dynamic loss function for neural network multi-objective training, optimizes dynamic loss with gradient-based HV maximization, and performs well in approximating the Pareto front. PHN-HVI (Hoang et al., 2023) employs a multi-sample hypernetwork and utilizes HV for hyperparameter optimization. PSL-HV1 and PSL-HV2 (Zhang et al., 2023) address the problem of PFL from the geometric perspective. Different from the method of this paper that focuses on calculating the gradient of HV, Zhang et al. (2023) adopt the polar coordinate system and utilize the R2 indicator to obtain the Pareto front. Despite the significance of the algorithms above, there is a room to improve the coverage of the entire Pareto front since they typically cluster around the middle and neglect the boundary solutions, and only identify partial solutions.

**Federated Learning** Federated Learning(FL) is a highly important paradigm of distributed machine learning that allows multiple FL participants (FL-PTs) to collaborate on training models without sharing their private data (Kairouz et al., 2021; Yang et al., 2020; Guo et al., 2025). This is particularly true in the era of foundation models (Ren et al., 2025; Fan et al., 2025; Yi et al., 2025; Wang et al., 2024). It is often viewed as a collaborative network of FL-PTs where a FL-PT can be complemented by other FL-PTs with different weights and a personalized model is trained for each individual FL-PT (Tan et al., 2022; Li et al., 2025; Miao et al., 2023; Williams, 2023). From some perspective, FL can be viewed as a type of multi-task learning, which is further a type of MOO (Sener & Koltun, 2018; Marfoq et al., 2021; Bai et al., 2020; 2024). There are $n$ FL-PTs, denoted by $\mathcal{V} = \{v_1, v_2, \ldots, v_n\}$; here $n$ is also the number of learning tasks. PFL can generate a Pareto front from which we can choose a $n$-dimensional preference vector $r^{i*}$ for each FL-PT $v_i$ such that the model performance of this FL-PT is the best when applying this vector. Then, $r^{i*}$ can be used to characterize the data complementarity between FL-PTs and quantifies the importance/weight of other FL-PTs $\mathcal{V} - \{v_i\}$ to $v_i$. All such preference vectors $\{r^{i*}\}_{i=1}^n$ can be used to form a directed weighted graph benefit graph $\mathcal{G}_b$ among FL-PTs, which has played a fundamental role in guiding the design of collabrative FL network in many use

cases (Cui et al., 2022; Wu & Yu, 2024; Ding & Wang, 2022; Bao et al., 2023; Pan et al., 2016; Tan et al., 2024; Chen et al., 2024; Li et al., 2025; Zhou et al., 2025). Some works assume that the values of $\{r^{i*}\}_{i=1}^{n}$ are known, and study how to determine a subgraph $\mathcal{G}_u$ of $\mathcal{G}_b$ that satisfies some desired properties to form stable coalitions, avoid conflicts of interests, or eliminate free riders. In $\mathcal{G}_u$, there is a direct edge from $v_j$ to $v_i$ if $v_j$ will make a contribution to $v_i$ in the actual FL training process, and it defines the collaborative FL network. The work of this paper aims to achieve a better coverage of the Pareto front and thus we can choose a more precise preference vector for each FL-PT. It can advance all the methodologies in these use cases when Hypernetworks is used to obtain the benefit graph.

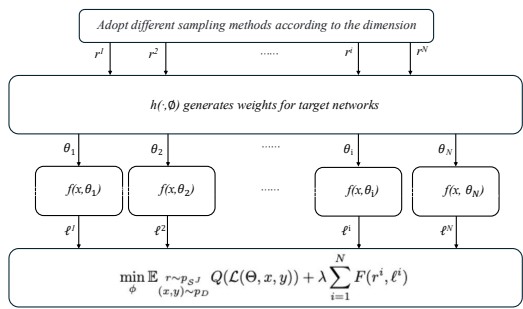

*Figure 1.* Multi-Sample Hypernetwork framework

## 3. Preliminary

The goal of Multi-task learning is to find $\theta^* \in \Theta$ to optimize $J$ loss functions (Sener & Koltun, 2018; Gupta et al., 2022):

$$\theta^* = \arg\min_{\theta} \mathbb{E}_{(x,y)\sim p_D} \ell(y, f(x;\theta)) \quad (1)$$

where $\ell(y, f(x;\theta)) = \{\ell_1(y, f(x;\theta)), \ldots, \ell_J(y, f(x;\theta))\}$, $p_D$ denotes the data distribution, $\ell_j : \mathcal{Y} \times \mathcal{Y} \to \mathbb{R}_{>0}$ denotes the $j$-th loss function and $f(x;\theta) : \mathcal{X} \times \vartheta \to \mathcal{Y}$ denotes the neural network with $\theta$. PFL extends this framework to learn the complete Pareto front. A hypernetwork is a deep model that generates the weights of the target network, limited by its input. Let $h(r; \phi)$ represents the hypernetwork with parameter $\phi$, and let $f$ represents the target network with parameter $\theta$, specially, $h(r; \phi)$ employs Multilayer Perceptron(MLP) to map the input preference vector $r$ to a higher dimensional space to construct shared features. These features create weight matrices for each layer in the target network through fully connected layers. We are trying to find the best parameter $\phi^*$:

$$\phi^* = \arg\min_{\phi \in \mathbb{R}^n} \mathbb{E}_{r \sim p_{SJ},(x,y)\sim p_D} F_e(\ell(y, f(x,\theta)), r) \quad (2)$$

where $F_e(\cdot, \cdot)$ is an extra criterion function such as LS or EPO function.

**Definition 3.1** (Dominance principle). For any $\theta_1, \theta_2$, we say that $\theta_1$ dominates $\theta_2$, denoted as $\theta_1 \prec \theta_2$, if and only if

$$\ell^i(y, f(x, \theta_1)) \leq \ell^i(y, f(x, \theta_2)), \forall i \in \{1, 2, \ldots, N\} \quad (3)$$

and $\ell(y, f(x, \theta_1)) \neq \ell(y, f(x, \theta_2))$.

**Definition 3.2** (Pareto set and Pareto front). The solution $\theta_i$ is called Pareto optimal when it satisfies: $\nexists \theta_j$ s.t. $\theta_j \prec \theta_i$. The Pareto set is defined as:

$$\mathcal{T} = \{\theta_i \in \Theta \mid \nexists \theta_j, \text{ s.t. } \theta_j \prec \theta_i\}. \quad (4)$$

The corresponding images in the objectives space are Pareto front $\mathcal{T}_f = \ell(y, f(x, \mathcal{T}))$.

HV is a commonly used metric to evaluate the quality of a $\mathcal{T}_f$. The HV of a set $A$ is the $J$-dimensional Lebesgue measure of the region dominated by $\mathcal{T}_f$ and bounded from above by reference point.

**Definition 3.3** (The hypervolume indicator). The HV indicator of a set $A$ is defined as:

$$\mathcal{H}_r(A) := \Lambda(\{a' \mid \exists a \in A : a \preceq a' \text{ and } a' \preceq \mathcal{R}\}) \quad (5)$$

where $\Lambda(\cdot)$ denotes the Lebesgue measure, and $\mathcal{R} \in \mathbb{R}^J$ is a reference point.

**Definition 3.4** (Voronoi Diagram). Let $\mathcal{X}$ be a metric space and $P = \{p_1, p_2, \ldots, p_N\}$ be a set of points in $\mathcal{X}$. The Voronoi grid $V(p_i)$ associated with a point $p_i$ is the set of all points in $\mathcal{X}$ that are closer to $p_i$ than to any other point in $P$. Formally, it is defined as:

$$V(p_i) = \{x \in \mathcal{X} \mid d(x, p_i) \leq d(x, p_j), \forall j \neq i\} \quad (6)$$

where $d(x, p_i)$ denotes the Euclidean distance between point $x$ and point $p_i$. The point $p_i$ is referred to as the site of Voronoi grid $V(p_i)$. The Voronoi Diagram is the union of all such grids, that is $V(P) = \bigcup_{i=1}^{N} V(p_i)$.

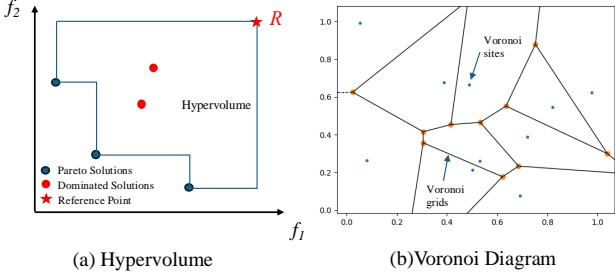

(a) Hypervolume        (b)Voronoi Diagram

*Figure 2.* Hypervolume (left) and Voronoi Diagram (right)

Figure 2(a) depicts the HV of the two-objective optimization problem, and Figure 2(b) depicts the Voronoi Diagram in the case of two-dimensional space.

# 4. Algorithm

## 4.1. Multi-Sample Hypernetwork

In this subsection, we introduce the existing challenge in multi-sample hypernetwork and propose a solution to solve it efficiently.

### 4.1.1. EXISTING CHALLENGE

We introduce the efficiency issue in multi-sample hypernetwork identified in (Hoang et al., 2023). Specifically, multi-sample Hypernetwork samples $N$ preference vectors $r^i$ and use $h(r; \phi)$ to generate $N$ target networks $f(x, \theta_i), i \in \{1, ..., N\}$ where $\theta_i = h(r^i, \phi)$. We need to evenly divide the space into $N$ sub-regions and collect vectors $r^i$ from these sub-regions. In $J$-dimensional space, the hyperplane $\mathcal{H}$ can be fully described as:

$$\mathcal{H} = \{(x_1, \ldots, x_J) \in \mathbb{R}^J \mid x_1 + \ldots + x_J = 1\} \quad (7)$$

where $1 \geq x_j \geq 0$, a partition with $N$ sub-regions should satisfy:

$$\bigcup_{i=1}^{N} \Omega^i = \mathcal{H} \text{ and } \Omega^{i'} \cap_{i' \neq i} \Omega^i = \emptyset \quad (8)$$

In the two-dimensional space, the polar coordinate is used for sampling to evenly divide the angle $\frac{i\pi}{2N}, i = 1, ..., N$. The high-dimensional uniform partitioning algorithm proposed by (Das & Dennis, 1998) encounters a significant challenge when dealing with points in $J$-dimensional ($J >= 3$) space: its computational complexity grows exponentially with the increase in dimensions, the number of points in the space $\mathbb{R}^J$ and $k = \frac{1}{\delta}$ is $\binom{J+k-1}{k}$, leading to an immense computational load that restricts the arbitrary setting of the number of rays.

### 4.1.2. SOLUTION: VORONOI SAMPLING

To decompose the high-dimensional design space $\mathcal{H}$ into separate regions and effectively cover the high-dimensional space with an arbitrary number of rays, we propose a genetic algorithm (GA)-based Monte Carlo Voronoi structure for high-dimensional region decomposition (Aurenhammer & Klein, 2000; Rubinstein & Kroese, 2016) as shown in Algorithm 1. The time complexity of Algorithm 1 is shown in Appendix A.1. Let $P = \{p_1, p_2, \ldots, p_N\}$ be the set of Voronoi site and $S = \{s_1, s_2, \ldots, s_M\}$ be the set of simulation points in $\mathcal{H}$. For each simulation point $s_m = (s_{m,1}, s_{m,2}, ..., s_{m,J}) \in S$, satisfing the constraint $\sum_{j=1}^{J} s_{mj} = 1$, we need to calculate the Euclidean distance $d(s_m, p_i)$ between $s_m$ and each site $p_i$, then we sort the distances $d(s_m, p_1), d(s_m, p_2), \ldots, d(s_m, p_N)$ for each $s_m$ in ascending order. The simulation point $s_m$ belongs to the grid of $p_k$ where $d(s_m, p_k)$ is the minimum among all

distances, i.e.,

$$s_m \in V(p_k) \text{ if } d(s_m, p_k) = \min_{i=1,...,N} d(s_m, p_i) \quad (9)$$

In order to improve the algorithm efficiency, we further introduce the KD tree (Bentley, 1975). Let $T$ be the KD-Tree constructed from the $P$, to find the nearest neighbor of a simulation point $s_m$, we use the KD-Tree search algorithm, which we denote as $NN(s_m, T)$, and the nearest point $p_k$ of $s_m$ using the KD-Tree can be written as: $p_k = NN(s_m, T)$. So Eq. (9) can be transformed into

$$s_m \in V(p_k) \text{ if } p_k = NN(s_m, T) \quad (10)$$

Here, based on the geometric distribution, we define uniformity as the situation where the number of points falling into each grid is equal. We use GA to find a more uniform partition of the space, the aim is to maximize the objective function $\mathcal{O}$. We set the objective function as follows:

$$\rho = \frac{1}{N} \sum_{i=1}^{N} (c(V(p_i)) - \overline{c})^2 \quad (11)$$

$$\mathcal{O} = 1/(1 + \rho) \quad (12)$$

where $c(V(p_i))$ is the count of points in grid $V(p_i)$ and $\overline{c}$ is the average quantity of points per grid. When $Var \to 0$, $\mathcal{O} \to 1$; and when $Var \to \infty$, $\mathcal{O} \to 0$. An ideal partition should satisfy Eq.(8) and the value of $\mathcal{O}$ in Eq.(12) is 1, where the number of points contained in each grid is equal. Algorithm 1 finally generates a fixed Voronoi partition of the hyperplane $\mathcal{H}$ which has the maximum value of $\mathcal{O}$. Figure 3 shows the grids generated by 3D Voronoi decomposition.

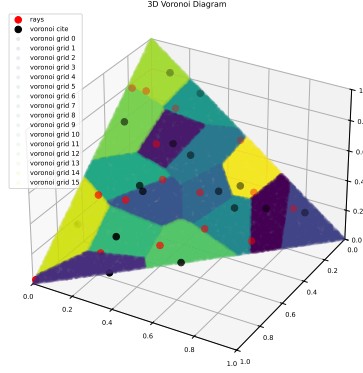

Figure 3. Voronoi Diagram with $J = 3, N = 16, M = 100000$

## 4.2. PHN-HVVS

The Voronoi Sampling method enables sampling rays across the entire $\mathcal{H}$. Building on this, we propose a new objective

---

**Algorithm 1** Voronoi Sampling

---

**Input:** the number of species $num\_species$, the number of points per species $N$, dimension $J$, Monte Carlo simulation number $M$ and $num\_generations$

**Initialize** population s.t. $\sum_{j=1}^{J} p_{ij} = 1, p_{ij} \in [0, 1]$.

**repeat**

Monte Carlo rand points set $S = \{s_1, s_2, ..., s_M\}$, where $s_m = (s_{m,1}, s_{m,2}, ..., s_{m,J}), \sum_{j=1}^{J} s_{mj} = 1$ in $\mathcal{H}$.

Voronoi Decomposition by Eq. (10).

Compute fitness by Eq. (12).

**Tournament selection**: Choose the individual with the highest fitness as the parent.

**Crossover**: $child = \alpha * parent_i + (1 - \alpha) * parent_j, \alpha \in [0, 1]$.

**Mutation**: $child = child + \epsilon, \epsilon \sim \mathcal{N}(0, (0.05)^2)$.

Update the fitness of the child.

Select the individual with the highest fitness as the basis for the next generation.

**until** $epoch > num\_generations$

Sample rays in the finally generated Voronoi grids.

---

function to better explore the solution space and achieve a complete Pareto front.

**P**areto **H**yper**N**etworks with **HV** maximization via **V**oronoi **S**ampling (PHN-HVVS) is designed to solve the following objective:

$$\min_{\phi} \mathbb{E}_{\substack{r \sim p_{\mathcal{S}^J} \\ (x,y) \sim p_D}} Q(\mathcal{L}(\Theta, x, y)) + \lambda \sum_{i=1}^{N} F(r^i, \ell^i) \quad (13)$$

where $\mathcal{L}(\Theta, x, y) = [\ell^1, ..., \ell^N]$, $Q = -HV(\mathcal{L}(\Theta, x, y))$, it satisfies $\nabla Q(\mathcal{L}(\Theta, x, y)) > 0$. $F$ is defined as $\mathcal{D}(r^i, \ell^i)$, $\mathcal{D}(r^i, \ell^i)$ represents the Euclidean distance from $\ell^i = (\ell_1^i, \ell_2^i, \ldots, \ell_J^i)$ to point $r^i = (r_1^i, r_2^i, \ldots, r_J^i)$ along the line with direction vector $\mathbf{u} = (1, \ldots, 1)_{1 \times J}$. The vector $\overrightarrow{r^i \ell^i} = \ell^i - r^i = (\ell_1^i - r_1^i, \ell_2^i - r_2^i, \ldots, \ell_J^i - r_J^i)$. We project the vector $\overrightarrow{r^i \ell^i}$ onto the direction vector $\mathbf{v}$, and the projection coefficient is $t$, where $t = \frac{\overrightarrow{r^i \ell^i} \cdot \mathbf{u}}{\mathbf{u} \cdot \mathbf{u}} = \frac{\sum_{j=1}^{J} (\ell_j^i - r_j^i) \cdot u_j}{\sum_{j=1}^{J} u_j^2}$. Eq. (14) is the distance from $r^i$ to the line which can be computed by the magnitude of the vector $\overrightarrow{r^i \ell^i} - t\mathbf{u}$.

$$\mathcal{D}(r^i, \ell^i) = \sqrt{\sum_{j=1}^{J} \left(\ell_j^i - \left(r_j^i + tu_j\right)\right)^2}$$
$$t = \frac{\sum_{j=1}^{J} \left(\ell_j^i - r_j^i\right) u_j}{\sum_{j=1}^{J} u_j^2} \quad (14)$$

Figure 4 depicts $\mathcal{D}(r^i, \ell^i)$ in 2D space.

The combination of HV and the penalty term optimizes both the quality and diversity of the Pareto front. If the HV of a set of solutions reaches its maximum, then these solutions are on the Pareto front (Fleischer, 2003). The HV term improves overall solution quality by pushing the front closer to the true Pareto front, while the penalty term $\left(\mathcal{D}\left(r^i, \ell^i\right)\right)$ ensures that the resulting distribution covers the entire Pareto front, regardless of its shape. This prevents solutions from concentrating in specific areas, a common issue in the convex part of Pareto optimization. Traditional MOEAs rely on the diversity of evolutionary population to search PF. The MOEA/D(Zhang & Li, 2007) and NSGA-III(Deb & Jain, 2013) variants can address convex problems by adopting adaptive reference vectors. Differently, PFL typically uses a hypernetwork to approximate the PF. Existing PFL methods employ gradient optimization to maximize HV by approximating gradients with HV contributions. However, as shown in (Zhang et al., 2023), convex PF boundary solutions suffer weight decay: intermediate solutions have larger HV gradients, causing preferential fitting of central regions. In traditional MOO, such weight decay does not need to be addressed, and MOEAs directly search boundary solutions through population diversity maintenance, without gradient reliance in PFL. Unlike cosine similarity, the distance-based penalty directly controls the geometric relationship between solutions and preferences, enabling the discovery of boundary-preference solutions to achieve complete coverage of the Pareto front. By working together, the HV term expands the front globally, and the distance-based penalty term refines its distribution, achieving a high-quality and diverse Pareto front. The entire algorithmic process is presented in Algorithm 2. More detail can be seen in Appendix A.4. Our goal is to optimize the sole parameter $\phi$ to find the Pareto front.

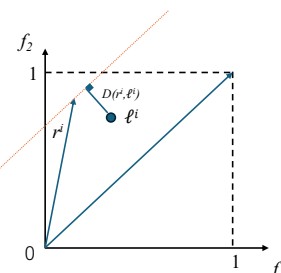

*Figure 4.* $\mathcal{D}(r^i, \ell^i)$ with the direction vector $v = (1, 1)$

## 5. Experiments

In this section, we will compare the performance of PHN-HVVS with other baselines PHN-HVI (Hoang et al., 2023), PHN-EPO, PHN-LS (Navon et al., 2020), COSMOS (Ruchte & Grabocka, 2021) and PHN-TCHE (Lin et al., 2022) in terms of the HV evaluation metric (Zitzler et al., 2007). This metric is crucial as it can comprehen-

**Algorithm 2** PHN-HVVS

1: **while** not converged do **do**
2:     **if** $J = 2$ **then**
3:        $r^1, \ldots, r^N \sim$ partition sample
4:     **else**
5:        $r^1, \ldots, r^N \sim$ Voronoi Sampling by Algorithm 1
6:     **end if**
7:     compute $\theta_i := h(r^i, \phi)$ for $i = 1, 2, \ldots, N$
8:     compute $g = -\sum_{i=1}^{N} \sum_{j=1}^{J} \frac{\partial HV(\mathcal{L}(\Theta; x, y))}{\partial \ell_j(\theta_i, f(x, \theta_i))} \frac{\partial \ell_j(\theta_i, f(x, \theta_i))}{\partial \phi}$
9:     $g_{\text{update}} := g - \lambda \sum_{i=1}^{N} \frac{\partial}{\partial \phi} \mathcal{D}\left(r^i, \ell^i\right)$
10:     $\phi \leftarrow \phi - \eta \cdot g_{\text{update}}$
11: **end while**
12: **return** $\phi$

sively reflect the quality of solutions in MOO problems. In high-dimensional space, HV can be effectively approximated(Bader & Zitzler, 2011), and there is a standard indicators .hv in Python's Pymoo library used extensively for compute HV. For the error of HV, when $J > 3$, the module in the library uses Monte Carlo method to sample about 10000 points to estimate the HV value. The error rate decreases with the square root of the sample size, and the actual error can be controlled within the range of 1% to 5%. In this paper, the value of this HV is only computed once and used for the final evaluation of algorithm performance. We do not calculate the specific value of HV every round, but instead use gradient descent method to obtain the $\phi$ that minimize Eq.(13). By analyzing their respective HV values, we can clearly understand the effectiveness of our approaches in obtaining a diverse and high quality set of solutions.

All experiments are trained on the NVIDIA GeForce RTX3090, the installed CUDA version is 12.2.

### 5.1. Toy Examples

For toy examples, the hypernetwork uses a 2-layer hidden MLP to generate the parameters of the target network, which consists of a single linear layer. For consistency, all methods share identical hyperparameters and run for the same number of iterations on the same problem.

**Problem 1 (Liu et al., 2021) with two Objective Functions**

$$\ell_1(\theta) = (\theta)^2, \ \ell_2(\theta) = (\theta - 1)^2, \ \text{s.t. } \theta \in \mathbb{R} \quad (15)$$

**Problem 2 (Lin et al., 2019) with two Objective Functions**

$$\ell_1(\theta) = 1 - \exp\left\{-\left\|\theta - \frac{1}{\sqrt{d}}\right\|_2^2\right\},$$
$$\ell_2(\theta) = 1 - \exp\left\{-\left\|\theta + \frac{1}{\sqrt{d}}\right\|_2^2\right\} \quad (16)$$
$$\text{s.t. } \theta \in \mathbb{R}^d, \ d = 100$$

**Problem 3 DTLZ2 (Zitzler et al., 2000) with three Objective Functions**

$$\ell_1(\theta) = \cos\left(\theta_1 \frac{\pi}{2}\right) \cos\left(\theta_2 \frac{\pi}{2}\right) \left(\sum_{i=3}^{10} (\theta_i - 0.5)^2 + 1\right),$$
$$\ell_2(\theta) = \cos\left(\theta_1 \frac{\pi}{2}\right) \sin\left(\theta_2 \frac{\pi}{2}\right) \left(\sum_{i=3}^{10} (\theta_i - 0.5)^2 + 1\right),$$
$$\ell_3(\theta) = \sin\left(\theta_1 \frac{\pi}{2}\right) \left(\sum_{i=3}^{10} (\theta_i - 0.5)^2 + 1\right),$$
$$\text{s.t. } \theta \in \mathbb{R}^{10}, \ 0 \le \theta_i \le 1$$
$$\quad (17)$$

**Problem 4 DTLZ4 (Zitzler et al., 2000) with three Objective Functions**

$$\ell_1(\theta) = \cos\left(\theta_1^\alpha \frac{\pi}{2}\right) \cos\left(\theta_2^\alpha \frac{\pi}{2}\right) \left(\sum_{i=3}^{10} (\theta_i^\alpha - 0.5)^2 + 1\right),$$
$$\ell_2(\theta) = \cos\left(\theta_1^\alpha \frac{\pi}{2}\right) \sin\left(\theta_2^\alpha \frac{\pi}{2}\right) \left(\sum_{i=3}^{10} (\theta_i^\alpha - 0.5)^2 + 1\right),$$
$$\ell_3(\theta) = \sin\left(\theta_1^\alpha \frac{\pi}{2}\right) \left(\sum_{i=3}^{10} (\theta_i^\alpha - 0.5)^2 + 1\right),$$
$$\text{s.t. } \theta \in \mathbb{R}^{10}, \ 0 \le \theta_i \le 1, \alpha = 100$$
$$\quad (18)$$

**Problem 5 ZDT1 (Zitzler et al., 2000) with two Objective Functions**

$$\min \ell_1(\theta_1) = \theta_1$$
$$\min \ell_2(\theta) = g\left(1 - \sqrt{\frac{\ell_1(\theta_1)}{g}}\right)$$
$$g(\theta) = 1 + 9\sum_{i=1}^{m} \frac{\theta_i}{(m-1)} \quad (19)$$
$$\text{s.t. } 0 \le \theta_i \le 1, \ i = 1, ..., m, m = 2$$

**Problem 6 ZDT2 (Zitzler et al., 2000) with two Objective**

**Functions**

$$\min \ell_1(\theta_1) = \theta_1$$
$$\min \ell_2(\theta) = g(\theta) \cdot h(\ell_1(\theta_1), g(\theta))$$
$$g(\theta) = 1 + \frac{9}{m-1} \cdot \sum_{i=2}^{m} \theta_i \qquad (20)$$
$$h(\ell_1(\theta_1), g(\theta)) = 1 - \left( \frac{\ell_1(\theta_1)}{g(\theta)} \right)^2$$
$$\text{s.t. } 0 \le \theta_i \le 1, i = 1, \dots, m, \quad m = 2$$

**Problem 7 VLMOP1 (Van Veldhuizen & Lamont, 1999) with two Objective Functions**

$$\ell_1(\theta) = \frac{1}{4n} \|\theta\|_2^2$$
$$\ell_2(\theta) = \frac{1}{4n} \|\theta - 2\|_2^2 \qquad (21)$$
$$\text{s.t.} n = 30$$

**Problem 8 VLMOP2 (Van Veldhuizen & Lamont, 1999) with two Objective Functions**

$$\ell_1(\theta) = \frac{1}{4n} \|\theta\|_2^2$$
$$\ell_2(\theta) = \frac{1}{4n} \|\theta - 2\|_2^2 \qquad (22)$$
$$\text{s.t.} -2 \le \theta_i \le 2, n = 10$$

The Pareto fronts of Problems 1, 5, and 7 are convex, while those of Problems 2, 3, 4, 6, and 8 are concave.

The HV values of the results can be found in Appendix B.3. By carefully observing the results presented in the figure, we can clearly see that the Pareto front obtained (denoted by the blue points) by our method, completely covers the entire true Pareto front. This demonstrates the robustness and adaptability of our approach in handling diverse Pareto front shapes. Compared to baseline methods, our approach achieves the best performance in terms of both diversity and quality of solutions.

It is obvious that PHN-HVI can effectively cover the concave portion of the Pareto front. Nevertheless, when it comes to the convex Pareto front, the solutions tend to be concentrated in the middle part of the front. PHN-EPO performs well in Problem 2 but finds the part of true Pareto Front in other Problems. It is well-recognized that PHN-LS effectively covers the convex part of the Pareto front. However, for concave Pareto fronts, such as in Problems 2, 3, 4, 6, and 8, it tends to gravitate toward the two ends (Boyd, 2004). The combined use of LS and cosine similarity in COSMOS leads to poor training performance when their optimization directions conflict. Similar to PHN-HVI, under mild conditions, PHN-TCHE aggregates in the middle region of the convex Pareto front.

## 5.2. Multi-Task Learning

**Datasets** We conduct experiments on datasets with different tasks. The datasets for the two tasks include MultiMNIST (MM.) (Sabour et al., 2017), Multi-Fashion (MF.), Multi-(Fashion+MNIST) (FM.), and Drug Review (Drug) Dataset (Gräßer et al., 2018). MM. is constructed by randomly selecting two images from the original MNIST (LeCun et al., 1998) dataset, placing one in the top-left corner and the other in the bottom-right corner of a new image. Each digit can be shifted by up to 4 pixels in any direction. Similarly, the MF. dataset is created by overlapping FashionMNIST items (Xiao et al., 2017), as well as FM. dataset by combining overlapping MNIST and FashionMNIST items. The training, validation, and testing ratio for each dataset is 11:1:2. The Drug Review dataset is examined by two tasks: (1) Prediction of the drug's rating through regression-based methods; (2) Classification of the patient's condition. The training, validation, and testing ratio for each dataset is 0.65:0.10:0.25.

The Jura dataset (Goovaerts, 1997) has 4 tasks. In the Jura dataset, the objective variables are zinc, cadmium, copper, and lead, representing 4 tasks. The dataset is divided into training, validation, and testing with a ratio of 0.65:0.15:0.20.

For higher-dimensional data, the SARCOS (SAR.) (Vijayakumar, 2000) dataset is used, which has 7 tasks. The SARCOS dataset aims to predict 7 joint torques, representing 7 tasks, from a 21-dimensional input space, which includes 7 joint locations, 7 joint velocities, and 7 joint accelerations. Additionally, 10% of the training data is set aside for validation.

**Model Architecture** We construct the hypernetwork for all datasets that employs a 2-layer hidden MLP to generate the parameters of the target network. For image classification, multi-LeNet (Sener & Koltun, 2018) is used as the target network. For text classification and regression, the target network is TextCNN (Rakhlin, 2016). The target network in Jura and SAR. is a Multi-Layer Perceptron with 4 hidden layers containing 256 units.

**Evaluation** Every method uses 25 preference vectors to evaluate each method. When dealing with two-task datasets, the reference point is set to (2, 2), and for multi-task datasets, it is set to $(1, \dots, 1)$ in accordance with the settings in (Hoang et al., 2023). Table 1 presents the HV values under the given reference point setting in the form of mean±standard deviation(std), which are the results of five independent runs.

The Pareto front of PHN-HVVS is widely dispersed. PHN-HVVS has a greater HV than previous methods. Our method has ability to effectively navigate the solution space and identify the Pareto Front. Additionally, our method allows

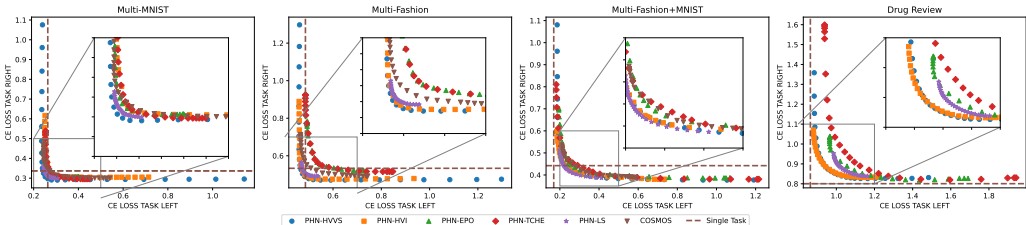

*Figure 5.* Results comparison on different Multi-Task Datasets

*Table 1.* Results compared to the state-of-the-art methods on Hypervolume.

|  | PHN-EPO | PHN-LS | PHN-TCHE | COSMOS | PHN-HVI | PHN-HVVS |
|---|---|---|---|---|---|---|
| MM. | 2.974±0.004188 | 2.976±0.007291 | 2.969±0.010703 | 2.977±0.014792 | 2.993±0.017194 | **3.008±0.016559** |
| MF. | 2.272±0.044656 | 2.283±0.030211 | 2.275±0.023285 | 2.274±0.039690 | 2.303±0.016967 | **2.328±0.035270** |
| FM. | 2.887±0.023533 | 2.881±0.027041 | 2.905±0.014980 | 2.857±0.019973 | 2.908±0.027066 | **2.947±0.020804** |
| Drug. | 1.194±0.005474 | 1.174±0.010247 | 1.206±0.004243 | NA | 1.308±0.081302 | **1.320±0.063629** |
| Jura | 0.876±0.041178 | 0.897±0.045139 | 0.883±0.044926 | 0.892±0.041809 | 0.922±0.044198 | **0.935±0.012841** |
| SAR. | 0.852±0.071599 | 0.858±0.070854 | 0.726±0.063612 | 0.865±0.068888 | 0.929±0.031036 | **0.939±0.026444** |

us to freely specify the number of rays regardless of the number of tasks, providing greater flexibility in handling complex optimization problems. In the drug review experiment, COSMOS fails to converge due to its mapping of the preference vector into an excessively high-dimensional embedding space. Figure 5 displays the results of one of the five runs, highlight the practical applicability of our method to complex multi-objective optimization tasks across various domains.

### 5.3. Federated Learning

In this subsection, we experimentally illustrate that the proposed method of this paper advances the methodologies of several problems in the FL field.

**Datasets and data heterogeneity** CIFAR-10 dataset (Krizhevsky et al., 2009), a benchmark for image classification, comprises 60,000 natural images evenly distributed across 10 object categories (6,000 samples per class). 10% of the training data are used for the validation split. The data heterogeneity is simulated via two typical approaches: pathological distribution (PAT.) (McMahan et al., 2017; Collins et al., 2021; Cui et al., 2022) and Dirichlet distribution (Dir.) (Tan et al., 2022; Ye et al., 2023), with $\beta = 0.5$. In particular, eICU (Pollard et al., 2018) is a dataset that gathers electronic health records (EHRs) from numerous hospitals across the United States for patients admitted to the intensive care unit (ICU). The objective is to forecast mortality during hospitalization. The training, validation, and testing sets are in a ratio of 0.7:0.15:0.15. The number of FL-PTs is set to 10.

**Benefit Graph** The benefit graph $\mathcal{G}_b$ characterizes the data

heterogeneity/complementarity between FL-PTs. Cui et al. (2022), Tan et al. (2024), and Chen et al. (2024) all use the hypernetwork technique to generate the benefit graph $\mathcal{G}_b$, and the effectiveness of using Hypernetwork to generate $\mathcal{G}_b$ is also validated in (Li et al., 2025). Specifically, there are $n$ FL-PTs. Each FL-PT $v_i$ has a local dataset $\hat{\mathcal{D}}_i$ and a risk/loss function $\ell^i \colon \mathbb{R}^n \to \mathbb{R}_+$. Given a learned hypothesis $q$, let the loss vector $\ell(q) = [\ell^1, \ldots, \ell^n]$ represent the utility loss of the $n$ FL-PTs under the hypothesis $q$. We say $q$ is a Pareto Solution if there is no hypothesis $q'$ that dominates $q$. Let $r^i = (r_1^i, \ldots, r_n^i) \sim Dir(\beta) \in \mathbb{R}^n$ denote a preference vector. Each vector represents the weight of the objective local model loss which is normalized with $\sum_{k=1}^n r_k^i = 1$ and $r_k^i \geq 0, \forall k \in \{1, \ldots, n\}$. The hypernetwork $HN$ takes $r$ as input and outputs a Pareto solution $q$, i.e.,

$$q \leftarrow HN(\phi, r), \tag{23}$$

where $\phi$ denotes the parameters of the hypernetwork. For each FL-PT $v_i$, linear scalarization can be used. An optimal preference vector $r^{i*} = \left( r_1^{i*}, r_2^{i*}, \ldots, r_n^{i*} \right)$ is determined to generate the hypothesis $q^{i*}$ that minimizes the loss with the data $\hat{\mathcal{D}}_i$. This is expressed as

$$q^{i*} = HN(\phi, r^{i*}) \text{ where } r^{i*} = \arg\max_{r^i} Per(HN(\phi, r^i)). \tag{24}$$

where $Per(HN(\phi, r^i))$ denotes the performance of $HN(\phi, r^i)$ on the validation data of $v_i$. For each FL-PT $v_i$, the value of $r_j^{i*}$ is used as an estimate to the weight of $v_j$ to $v_i$. $\{r^{i*}\}_{i=1}^n$ defines a directed weighted graph, i.e., the benefit graph $\mathcal{G}_b$.

Based on the number of FL-PTs, which corresponds to the dimension of the objective functions, different sampling

methods are adopted according to this paper. Meanwhile, the objectives are transformed into Eq. (13). More details can be seen in Appendix A.5.

**Metrics** MTA represents the mean test accuracy. We use MTA to measure the performance of different methods on CIFAR-10. To address the severe class imbalance in the eICU dataset (negative samples over 90%), we adopt the AUC (Area Under the ROC Curve) metric for performance evaluation, as it is statistically robust to skewed label distributions. The results are presented in the form of mean ± std under five runs.

In Table 2, we carry out experiments based on the work of Cui et al. (2022). We can see that better results are obtained by adding our method.

*Table 2.* Performance comparison on CIFAR-10 under different data heterogeneity (MTA, CE).

|  | CE | .+HVVS |
| --- | --- | --- |
| PAT. | 74.38±4.00 | **86.88±2.57** |
| Dir($\beta = 0.5$) | 47.13±2.03 | **58.35±2.37** |

*Table 3.* Performance comparison on CIFAR-10 under different data heterogeneity (MTA, FedCompetitors).

|  | FedCompetitors | .+HVVS |
| --- | --- | --- |
| PAT. | 82.19 ± 0.41 | **82.44 ± 0.35** |
| Dir($\beta = 0.5$) | 47.96 ± 0.75 | **51.72 ± 1.39** |

*Table 4.* Performance comparison on eICU (AUC, FedEgoists).

|  | FedEgoists | .+HVVS |
| --- | --- | --- |
| $v_0$ | **66.36±19.28** | 66.12±8.45 |
| $v_1$ | 81.58±6.65 | **86.04±3.41** |
| $v_2$ | 66.04±33.21 | **71.97±6.54** |
| $v_3$ | **84.40±5.76** | 75.39±10.68 |
| $v_4$ | 75.84±11.26 | **82.89±6.79** |
| $v_5$ | **68.41±5.60** | 66.80±8.14 |
| $v_6$ | 56.86±7.52 | **69.80±4.96** |
| $v_7$ | 77.97±14.94 | **91.77±2.16** |
| $v_8$ | 90.60±10.57 | **100.00±0.00** |
| $v_9$ | 79.88±8.29 | **87.18±5.08** |
| Avg | 74.79 | **79.80** |

FedCompetitors (Tan et al., 2022) and FedEgoists (Chen et al., 2024) considered the competing relationships between FL-PTs or the self-interest of each FL-PT. The competition rate in the experiment of FedCompetitors is set to 0.2. We also test the performance of FedEgoist under different competition rates and different data distributions. Overall, PHN-HVVS plays a fundamental role and integrating

*Table 5.* Performance comparison on CIFAR-10 under Dirichlet distribution ($\beta = 0.5$) with different compete ratio (MTA, FedEgoists).

| Dir($\beta = 0.5$) | FedEgoists | .+HVVS |
| --- | --- | --- |
| 0.05 | 62.16±0.72 | **62.94±0.97** |
| 0.1 | 61.93±0.92 | **63.59±1.08** |
| 0.2 | 62.39±1.26 | **63.78±1.16** |
| 0.3 | 62.47±0.59 | **62.52±1.95** |
| 0.4 | 61.21±0.80 | **62.00±2.40** |

*Table 6.* Performance comparison on CIFAR-10 under pathological distribution with different compete ratio (MTA, FedEgoists).

| PAT. | FedEgoists | .+HVVS |
| --- | --- | --- |
| 0.05 | **80.60±1.67** | 80.41±0.97 |
| 0.1 | 80.87±1.01 | **81.16±0.70** |
| 0.2 | 80.85±1.07 | **81.76±0.53** |
| 0.3 | 81.49±1.46 | **81.53±2.09** |
| 0.4 | 81.37±1.01 | **81.63±2.29** |

PHN-HVVS into these methodologies in FL has brought significant improvements.

## 6. Conclusion

In this paper, we propose the PHN-HVVS algorithm that leverages Voronoi Sampling in high-dimensional spaces. Additionally, we introduce a novel objective function to tackle the PFL problems. Our approach is capable of delving deeper into the Pareto front space, thereby obtaining a broader range of Pareto front that can better approximate the true front. This provides a more accurate representation of the optimal trade-offs in the PFL problems. Moreover, we apply the proposed method to federated learning. Extensive experiments show that the proposed method can adaptively work well under federated learning setting. This work can inspire further developments in the application of MOO techniques to federated learning.

## Acknowledgements

This work is supported in part by the National Key R&D Program of China (Grant No. 2024YFE0200500); the National Natural Science Foundation of China (Grant Nos. 62327801 and 62302147); the International Collaboration Fund for Creative Research of the National Science Foundation of China (NSFC ICFCRT) (Grant No. W2441019); the Ministry of Education, Singapore, under its Academic Research Fund Tier 1 (RG101/24); the National Research Foundation, Singapore, and DSO National Laboratories under the AI Singapore Programme (AISG Award No. AISG2RP-2020-019);

and the MTI, Singapore, under its AI Centre of Excellence for Manufacturing (AIMfg) (Award W25MCMF014).

## Impact Statement

This paper presents work whose goal is to advance the field of Machine Learning. There are many potential societal consequences of our work, none which we feel must be specifically highlighted here.

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

# A. Theoretical Analysis.

## A.1. The Time Complexity of Algorithm 1

Initializing the population involves assigning values to an array of size $N \times J$. Since there are $num\_species$ species in total, the time complexity is $O(num\_species \times N \times J)$. The time complexity of generating $M$ random points through Monte Carlo simulation is $O(M \times J)$. When performing Voronoi decomposition, the complexity of building a KD-tree is $O(N \log(N))$, and the complexity of querying the nearest neighbors is $O(M \log(N))$. Therefore, the total time complexity of decomposition is $O((N+M) \log(N))$. The time complexity of calculating the fitness is $O(num\_species \times N)$. The time complexity of crossover and mutation is $O(num\_species \times N \times J)$. Since the main loop is executed $num\_generations$ times, the overall time complexity of the Algorithm 1 is $O(num\_generations \times (N + M) \log(N))$.

## A.2. The details of Algorithm 1

Algorithm 1 uses Monte Carlo simulation to generate points located on hyperplane $\mathcal{H}$ in each round, and then searches for the nearest Voronoi site $p_n$ for these $M$ points. The genetic algorithm is applied to optimize the objective function in Eq.(12). In genetic algorithm, we randomly select three individuals and then choose the optimal individual from them. The crossover rate $\alpha$ is uniformly distributed within the range of [0,1]. The mutation method is to randomly perturb a certain point, and the range of the mutation is controlled by the parameter mutation_std. In this article, mutation_std is 0.05. When a newly generated point exceeds the valid range [0,1] in any dimension, the algorithm calculates a scaling factor to project the point onto the nearest boundary while preserving its original direction, provided that the directional variation component in that dimension is non-zero. This ensures that the mutated points are within a reasonable range of values.

## A.3. The relationships between Algorithm 1 and 2

Algorithm 1 optimizes the objective function in Eq.(12) using a genetic algorithm, ultimately yielding $N$ fixed Voronoi grids. In each grid, the Voronoi sites $P = \{p_1, p_2, \ldots, p_N\}$ and simulation point set $S = \{s_1, s_2, \ldots, s_M\}$ are stored, with each $s_m \in S$ assigned a label indicating its partition. Algorithm 2 calls Algorithm 1 at line 5 and directly leverages the generated Voronoi grid structure thereby to perform fast sampling in each round within these grids (simulation point sets with the same label) without requiring recalculation. Each grid $V(p_i)$ is equivalent to a subregion $\Omega^i$. As detailed in (Hoang et al., 2023), adopting partition can enhance the performance of the HV and penalty functions. The round-by-round sampling in the fixed Voronoi grids aims to explore the entire space and obtain a complete PF. The Voronoi partition guarantees the coverage of each region during the sampling process, facilitating efficient and comprehensive exploration.

## A.4. The process of HvMaximization

The HvMaximization is a MOO algorithm that calculates dynamic weights for optimizing HV using a method based on HIGA-MO (Wang et al., 2017). The operation begins by performing non-dominated sorting (Deb et al., 2002) on the provided multi-objective solutions to create multiple Pareto fronts. Next, a dynamic reference point is calculated by scaling the maximum values of the objectives with a factor of 1.1. The dynamic reference point is updated to ensure that it remains outside the current solution space. Once the fronts are identified and the reference point is adjusted, the algorithm calculates the HV gradients, which represent the rate of change in HV with respect to each solution in the front, through Algorithm 3 (Emmerich & Deutz, 2014). To ensure consistency in the objective space, the gradients are normalized. If any gradient has a norm close to zero, it is adjusted to avoid division by zero errors. After normalization, the gradients are used to compute dynamic weights for each solution, which reflect how much each solution contributes to maximizing the overall HV. The overall goal is to maximize the HV by assigning appropriate weights to each solution, allowing for better exploration and exploitation of the solution space. Algorithm 3 illustrates the detailed steps of the GRADMULTISWEEP function.

## A.5. The generation of benefit graph $\mathcal{G}_b$

In FL, the $n$ FL-PTs correspond to $n$ learning tasks. The heterogeneity of data across FL-PTs entails evaluating the complementarity of data between FL-PTs, where a benefit graph arises. In multiple works (Cui et al., 2022; Tan et al., 2024; Chen et al., 2024), the benefit graph is assumed to be known, and estimated by PFL schemes. Optimization process is further taken in these existing FL algorithms for different purposes. The preciseness of these parameters directly affects the performance of these FL algorithms where the PFL scheme used previously is the one (i.e., PHN-LS) in (Navon et al., 2020). However, it can also be the scheme (i.e., PHN-HVVS) proposed in this paper. With PHN-HVVS, all these previous FL

---

**Algorithm 3** GRADMULTISWEEP

---

**Require:** Vector $\mathbf{Y}$ with subvectors, reference point $\mathcal{R}$
**Ensure:** Partial derivatives for each subvector component
 1: Divide subvectors into sets $Z, U, E$
 2: **if** $U \neq \emptyset$ **then**
 3:     **output** One-sided derivatives in $U$
 4: **end if**
 5: Set derivatives of $Z$ subvectors to 0
 6: Compute derivatives for $E$ subvectors
 7: **for** each dimension $j$ **do**
 8:     Project $P$ subvectors to $(J-1)$-dim, omit $j$-th coordinate.
 9:     Sort projected subvectors by $j$-th coordinate, add to queue $Q$
10:     Initialize tree $T$ for non-dominated points
11:     **while** $Q$ is not empty **do**
12:         Remove max element $q$ from $Q$
13:         Compute HV change $\Delta H(q, T)$
14:         Set derivative $\frac{\partial \mathcal{H}}{\partial y^{(i(q))}} = \Delta H(q, T)$, where i(q) is the index that corresponds to the index of the original subvector in $\mathbf{Y}$ of which q is the projection.
15:         Add $q$ to $T$, remove Pareto dominated points
16:     **end while**
17: **end for**

---

algorithms achieve a better performance since the PF is better covered.

Below, we introduce the three relevant works. First, Cui et al. (2022) study the collaboration equilibrium in FL where FL-PTs are divided into multiple groups or coalitions. Let $\pi(i)$ denote the unique coalition to which FL-PT $i$ belongs. Cui et al. (2022) study how to form a core-stable coalition structure $\pi$, which guarantees that there is no coalition $\mathcal{C}$ such that every FL-PT $i \in \mathcal{C}$ prefers $\mathcal{C}$ over $\pi(i)$. Only FL-PTs within the same coalition contribute to each other, forming a subgraph of $\mathcal{G}_b$. Second, in cross-silo FL, FL-PTs are typically organizations. FL-PTs in the same market area may compete while those in different areas are independent. Tan et al. (2024) extend the principle "the friend of my enemy is my enemy" to prevent FL-PTs from benefiting their enemies in collaborative FL, forming a subgraph of $\mathcal{G}_b$. Third, Chen et al. (2024) address self-interested FL-PTs in cross-silo FL and propose a framework to simultaneously eliminate free riders (who benefit without contributing) and avoid conflict of interest between FL-PTs, also forming a subgraph of $\mathcal{G}_b$.

In this paper, we employ PHN-HVVS to generate the benefit graph $\mathcal{G}_b$, introducing modifications to the sampling distribution and solver method. Initially, preference vectors of equal length were generated according to the number of FL-PTs. These preference vectors could follow a uniform distribution, Dirichlet distribution, or the novel Voronoi distribution proposed in this study. Subsequently, these preference vectors served as inputs to the hypernetwork, which then generated the weights for the target network. The target network processed image features to produce prediction results, which were compared with the ground-truth labels to compute the loss. Following this, solvers, including the LS and the newly proposed HVVS in this paper, were utilized to address the total loss and conduct gradient optimization. In contrast to traditional FL algorithms, this approach innovatively integrates the hypernetwork architecture with multi-objective optimization, thereby offering a novel solution for generating the benefit graph.

## B. Experimental Details

### B.1. The notation table

To improve the readability of our paper, we've provided a summary of key notations in Table 7.

### B.2. The objective of baselines

**PHN-EPO** The PHN-EPO optimizes $\min_\phi \mathbb{E}_{r \sim p_{\mathcal{S}^J}, (x,y) \sim p_D} EPO(\ell(y, f(x, \theta)), r)$, the expected value of EPO loss. PHN-EPO requires solving each EPO subproblem.

*Table 7.* The notation table.

| Variable | Definition |
|---|---|
| $\mathcal{V}$ | The set of FL-PTs |
| $v_i$ | The $i$-th FL-PT |
| $r$ | The preference vector/ray |
| $\mathcal{R}$ | The reference point |
| $n$ | The number of FL-PTs |
| $\phi$ | The parameter of hypernetwork |
| $\theta$ | The parameter of target network |
| $\ell(\cdot, \cdot)$ | The loss function |
| $h(r; \phi)/HN(\phi; r)$ | The hypernetwork |
| $f(x; \theta)$ | The target network |
| $d(\cdot, \cdot)$ | The Euclidean distance between different points |
| $\mathbf{u}$ | The direction vector $(1, \ldots, 1)_{1 \times J}$ |
| $D(r^i, l^i)$ | The Euclidean distance from $\ell^i$ to point $r^i$ along the line with direction vector $\mathbf{u}$ |
| $P$ | The set of Voronoi sites |
| $S$ | The set of simulation points |
| $p_i$ | The $i$-th Voronoi site |
| $s_m$ | The $m$-th simulation points |
| $N$ | The number of Voronoi grids |
| $M$ | The number of simulation points |
| $J$ | The number of objectives |
| $\mathcal{T}$ | The Pareto front |
| $T$ | The KD-Tree |
| $\Omega^i/V(p_i)$ | The Voronoi grid with the Voronoi site $p_i$ |
| $q$ | The hypothesis/The pareto solution |

**PHN-LS** The PHN-LS optimizes $\min_\phi \mathbb{E}_{r \sim p_{\mathcal{S}^J}, (x,y) \sim p_D} \sum(\ell(y, f(x, \theta))r)$, this method can not find the concave part of the Pareto front.

**PHN-TCHE** The PHN-TCHE optimizes $\min_\phi \mathbb{E}_{r \sim p_{\mathcal{S}^J}, (x,y) \sim p_D} \max(\ell(y, f(x, \theta)), r)$, this method can not find the concave part of the Pareto front.

**COSMOS** The COSMOS optimizes $\min_\phi \mathbb{E}_{r \sim p_{\mathcal{S}^J}, (x,y) \sim p_D} \sum(\ell(y, f(x, \theta))r) - \lambda \cos(r, \mathcal{L})$, this method combines LS with consine similarity.

**PHN-HVI** The PHN-HVI optimizes $\min_\phi -\mathbb{E}_{r^i \sim p_{\mathcal{S}^J}, (x,y) \sim p_D} \text{HV}(\mathcal{L}(\Theta; x, y)) + \lambda \sum_{i=i}^p \cos(\vec{r}^i, \vec{\ell^i})$, this method can not find the convex part of the Pareto front.

### B.3. The HyperVolume of Toy Examples

The reference point for the two objectives-functions is (2,2), for the three objectives-functions is (2,2,2). A higher HV indicates a better-quality Pareto front. The Pareto front generated by our method exhibits the largest HV among all the compared baselines. This remarkable outcome not only clearly shows its ability to accurately approximate the true Pareto front but also vividly demonstrates its superiority in MOO. The results are presented in Table 8 and Figures 10-16, where the table reports the mean and standard deviation across five independent runs, while the figures illustrate the outcomes of one run among these five.

### B.4. Different sampling techniques comparison

We conduct extended experiments on the Jura and SARCOS datasets, comparing our proposed method against multiple naive sampling techniques, including: Uniform random sampling, Latin hypercube sampling, Polar coordinate sampling, Dirichlet distribution sampling, K-means clustering-based representative selection. The sampling points within each Voronoi region are closest to the site of that region, which naturally avoids the problem of local aggregation or omission in the

*Table 8.* Results comparison on different problems.

| | PHN-EPO | PHN-LS | PHN-TCHE | COSMOS | PHN-HVI | PHN-HVVS |
|---|---|---|---|---|---|---|
| Pro.1 | 3.790±0.007230 | 3.832±0.000526 | 3.813±0.005223 | 3.210±0.736292 | 3.791±0.002538 | **3.833±0.000026** |
| Pro.2 | 3.304±0.063656 | 3.362±0.016258 | 3.321±0.086994 | 3.344±0.043419 | 3.374±0.011347 | **3.387±0.007168** |
| Pro.3(DTLZ2) | 7.299±0.010926 | 6.015±0.084161 | 7.303±0.011461 | 7.345±0.019812 | 7.385±0.006708 | **7.388±0.001225** |
| Pro.4(DTLZ4) | 7.380±0.010358 | 6.020±1.096984 | 7.267±0.010933 | 7.236±0.010072 | 7.383±0.005471 | **7.386±0.003612** |
| Pro.5(ZDT1) | 3.651±0.075391 | 3.654±0.085641 | NA | 3.631±0.036321 | 3.520±0.012252 | **3.735±0.001754** |
| Pro.6(ZDT2) | 2.047±1.094539 | 2.0±0 | 2.800±0.652913 | 3.057±0.528525 | 3.321±0.022386 | **3.323±0.004026** |
| Pro.7(VLMOP1) | 3.810±0.003677 | 3.825±0.010152 | 3.816±0.006643 | 3.747±0.012842 | 3.782±0.000523 | **3.829±0.000629** |
| Pro.8(VLMOP2) | 3.308±0.020874 | 3.313±0.021368 | 3.329±0.012204 | 3.299±0.014255 | 3.335±0.001013 | **3.339±0.000001** |

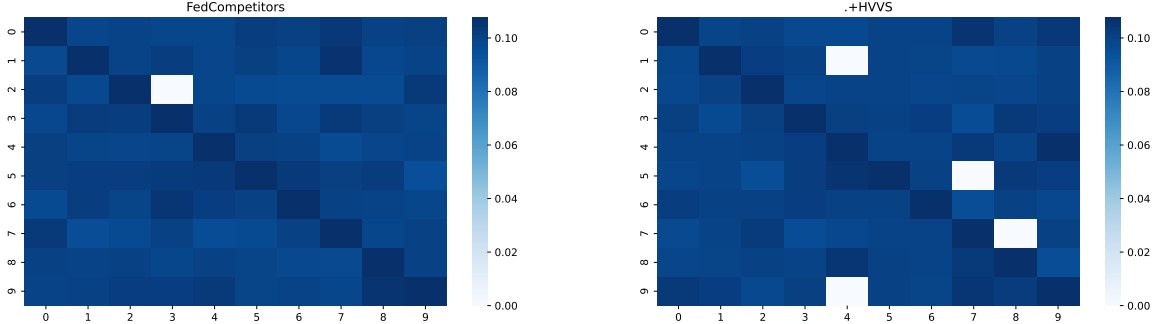

*Figure 6.* Pathological distribution Benefit Graph

sampling distribution. Voronoi partition method ensures effective exploration of the global PF. As shown in Table 9, the proposed approach outperforms other strategies.

*Table 9.* Comparison of different sampling techniques on Jura and SARCOS

| | Random | Latin | Polar | Dir. | K-means | Voronoi |
|---|---|---|---|---|---|---|
| Jura | 0.928 | 0.922 | 0.928 | 0.923 | 0.925 | **0.935** |
| SARCOS | 0.884 | 0.883 | 0.881 | 0.888 | 0.877 | **0.949** |

## B.5. The details of the federated learning experiment

For the CIFAR-10 dataset, the hypernetwork leverages a 2-layer hidden MLP to generate the parameters of the target network (He et al., 2024; 2021; Shang et al., 2023; Han et al., 2023; Wu et al., 2019b; 2021; Wu & Loiseau, 2023; 2024). The target network is the same as (Cui et al., 2022; Tan et al., 2024; Chen et al., 2024) and the reference point is set as $(3, ..., 3_J)$. For the eICU dataset, we construct a single-layer hidden MLP hypernetwork. The target network utilizes a Transformer classifier with Layer Normalization. The reference point is set as $(1, ..., 1_J)$. We integrate the proposed method in this paper into CE, FedCompetitors and FedEgoists. As can be seen from the table 2-6, the proposed method has brought about some degree of improvement.

In the experiment of FedCompetitors, we set up a total of 10 FL-PTs. In the Pathological distribution, we randomly assign two categories of CIFAR-10 (5,000 images in each category) to each FL-PT. In the Dirichlet distribution, we set $\beta = 0.5$ and the distribution vector is drawn from $\text{Dir}(\beta)$ for each FL-PT. We can get the Benefit Graph as shown in Figure 6 and 7. We set the competition rate of the competition matrix to 0.2. This means that the probability of any two FL-PTs being competitors is 0.2. We randomly generate the competition graph in five runs and then observe the performance of the algorithm. At a certain run, the competing FL-PTs are $(v_0, v_5), (v_1, v_2), (v_2, v_4), (v_3, v_4), (v_3, v_6), (v_4, v_6), (v_4, v_7), (v_4, v_9)$ in the Pathological distribution case. The accuracy of FedCompetitors is 82.15 and the accuracy of.+HVVS is 82.23. This is illustrated in Figure 8. In the Dirichlet distribution case, the competing FL-PTs are $(v_0, v_2), (v_1, v_2), (v_1, v_8), (v_2, v_5), (v_2, v_7), (v_3, v_4), (v_3, v_5), (v_6, v_8)$. The accuracy of

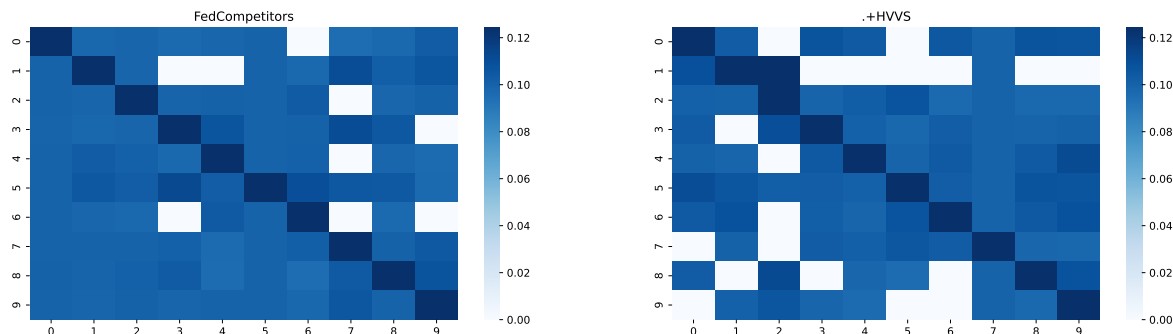

*Figure 7.* Dirichlet distribution Benefit Graph

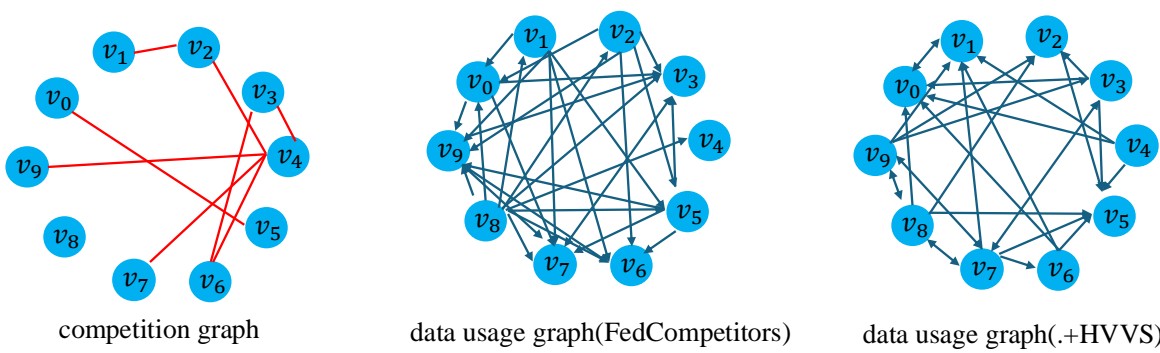

competition graph                    data usage graph(FedCompetitors)                    data usage graph(.+HVVS)

*Figure 8.* The Competition Graph and Data Usage Graph under Pathological distribution

FedCompetitors is 48.28 and the accuracy of.+HVVS is 53.54. This is illustrated in Figure 9. We can find that the proposed method application can better identify the relationships that align with the actual situations of FL-PTs, thus leading to better experimental results.

## B.6. Computer Resources

We utilize 8 NVIDIA GeForce RTX 3090 with 24GB of memory. The installed CUDA version is 12.2, and the graphics driver version is 535.104.05.

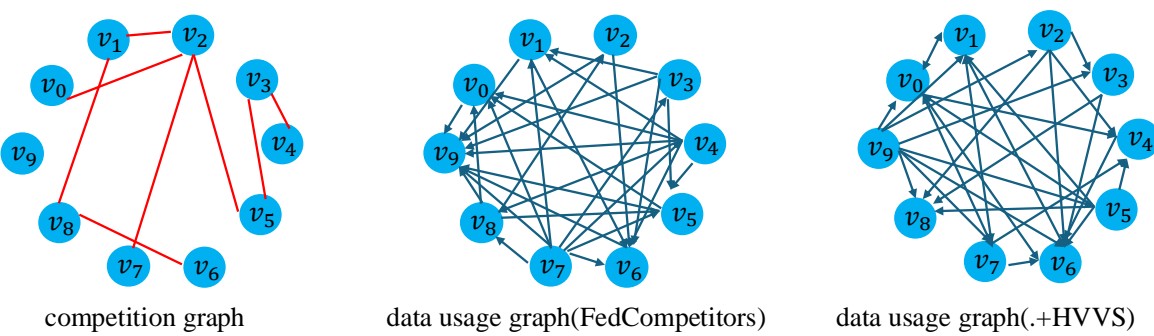

competition graph                    data usage graph(FedCompetitors)                    data usage graph(.+HVVS)

*Figure 9.* The Competition Graph and Data Usage Graph under Dirichlet distribution

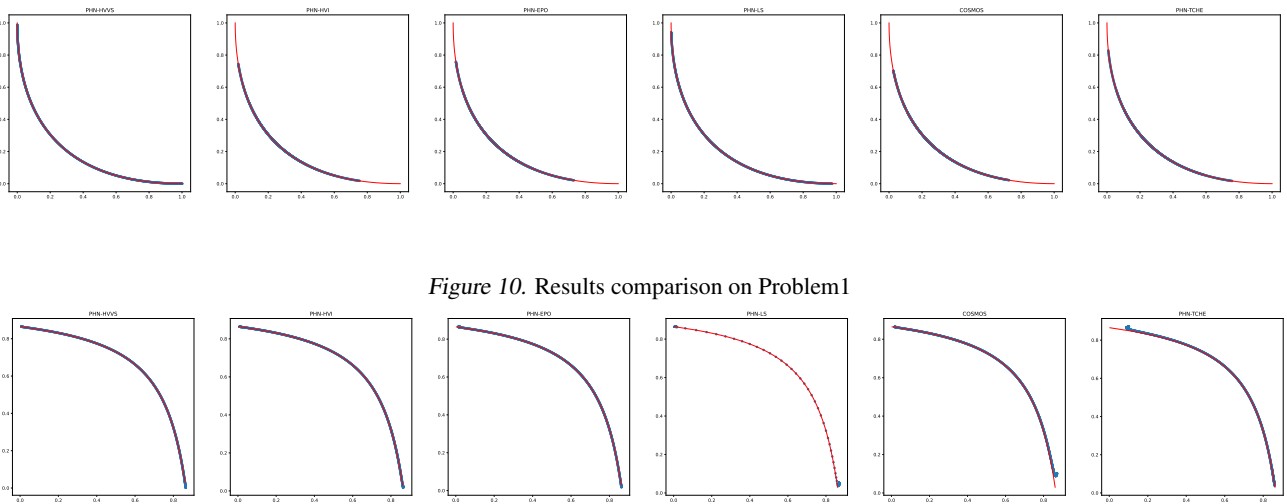

*Figure 10.* Results comparison on Problem1

*Figure 11.* Results comparison on Problem2

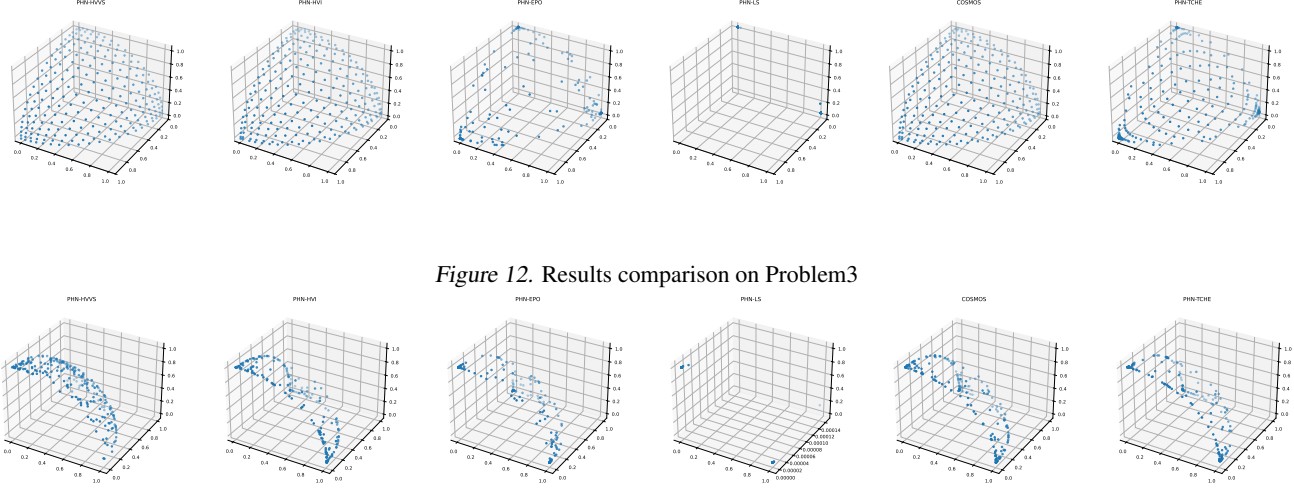

*Figure 12.* Results comparison on Problem3

*Figure 13.* Results comparison on Problem4

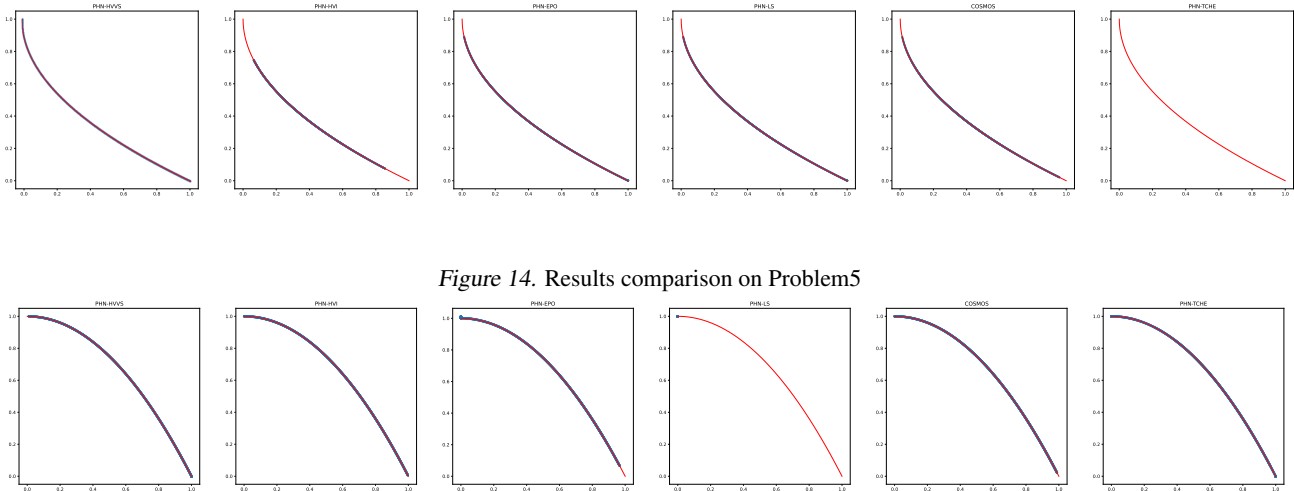

*Figure 14.* Results comparison on Problem5

*Figure 15.* Results comparison on Problem6

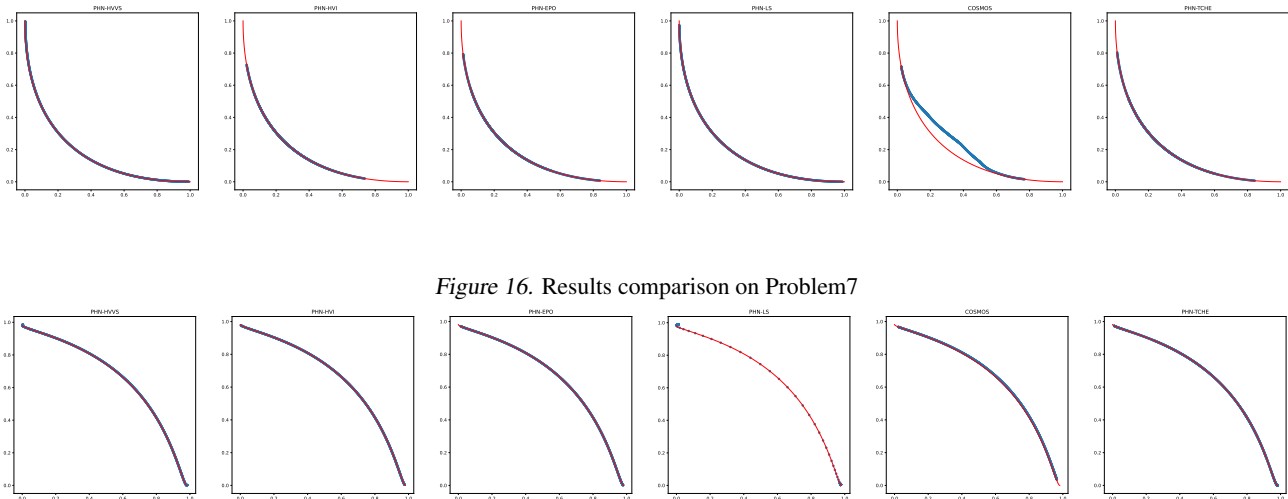

*Figure 16.* Results comparison on Problem7

*Figure 17.* Results comparison on Problem8

