# OpenReview forum: "Voronoi-grid-based Pareto Front Learning and Its Application to Collaborative Federated Learning"
_ICML.cc/2025/Conference — ICML 2025 poster_

### Official Review · Reviewer_J3SQ · 2025-03-02

**Overall Recommendation:** 5

**Summary:**

This paper studies an interesting and important question, which is about the use of hypernetworks to efficiently approximate the Pareto Front. The proposed approach, PHN-HVVS, addresses multi-objective optimization (MOO) tasks in machine learning by novelly designing a novel loss function and sampling rays from Voronoi grids in high-dimensional spaces. The authors validated the effectiveness of the proposed solution on multiple typical examples, benchmark datasets and real-world datasets. The proposed approach of this paper has a fundamental role in many applications, as illustrated in the experimental part.

**Claims And Evidence:**

yes

**Essential References Not Discussed:**

The references of this paper are sufficient.

**Experimental Designs Or Analyses:**

The authors took extensive experiments on various applications to validate the effectiveness of the proposed solution. These experiments also show the fundamental role of the proposed approach in many applications. In addition to what is traditionally done in the experimental part, the authors also show a new recent application of the proposed approach to federated learning, which advances the methodologies of several promising problems in the recent papers published at other major venues such as NeurIPS, KDD and AAAI. In addition, to help show the reproducibility of the proposed solution, the authors can make their code available publicly or give more details on the design of the target neural network structure.

**Methods And Evaluation Criteria:**

Yes, the proposed methods and evaluation criteria make sense. Like what is commonly done, the authors use Hypervolume as a metric to measure the quality of a set of solutions in the Pareto front; here, a larger hypervolume indicates a better performing set of solutions. The authors explore high-dimensional sampling by novelly using Voronoi grid and designing a new penalty term; here, the integration of Voronoi grid with genetic algorithms addresses sampling rays in high-dimensional spaces effectively and the novel loss function design balances Hypervolume optimization and Pareto front diversity. Pareto front learning in high-dimensional spaces is of high importance to many applications, e.g., in federated learning, it is important to have the number of objectives to be up to 10.

**Other Comments Or Suggestions:**

Somethings can be done to help non-expert readers better understand your work. Specifically, the authors can give more details about Algorithm 1, such as the number of parents involved in the tournament selection, the mutation rate, and how the mutation steps are handled if the offspring exceed the predefined parameter range.

**Other Strengths And Weaknesses:**

Strengths:
	The paper is well organized and easy to follow.
	The overall framework is novel. The integration of Voronoi grid with genetic algorithms addresses sampling rays in high-dimensional spaces effectively and the novel loss function design balances Hypervolume optimization and Pareto front diversity.
	Pareto front learning in high dimensional spaces is important. The research question of PFL has potential practical impact in real-world settings.
Weaknesses:
	Please refer to the part “Other Comments Or Suggestions” below

**Questions For Authors:**

Please refer to the above part ”Other Comments or Suggestions”

**Relation To Broader Scientific Literature:**

This paper is related to two promising research topics, Pareto-front learning and federated learning, respectively. It is closely tied to the broader scientific literature (Okabe et al., 2004; Navon et al., 2020; Hoang et al., 2023; Cui et al., 2022; Tan et al., 2024; Chen et al., 2024). For example, Okabe et al. (2004) introduced VEDA, which laid the foundation for Voronoi-based methods in the field of multi-objective optimization (MOO). Navon et al. (2020) proposed the concept of Pareto-front learning (PFL) and Pareto HyperNetworks (PHNs) to address the MOO problems for machine learning tasks. Hoang et al. (2023) contributed to the understanding of multi-sample hypernetworks and hypervolume in PFL, however, their PHN-HVI scheme has limitations in covering the convex part of the Pareto front, which is significantly improved by this paper. Federated learning allows multiple data owners to collaboratively train machine learning models in a privacy-preserving way. Besides what have been done by these previous works, the authors of this paper also highlight a new application of PFL to federated learning where PFL is the foundation of the methodologies of several recent problems in federated learning, and PFL can be used to evaluate how important a data owner is to the other data owners. In many cases, such evaluation is the basis to optimize the collaboration relationships among data owners in multiple promising works at major venues in KDD’22, NeurIPS’24, and AAAI’24 (Cui et al., 2022; Tan et al., 2024; Chen et al., 2024).

**Theoretical Claims:**

Yes, I checked. Eq. (14) defines the distance metric between the solution and the preference vector along the given direction. Although it is correct, the authors can explain this equation a bit more.

---

> ### Author Rebuttal · Authors · 2025-04-01
>
> **Comments 1**. Eq. (14) defines the distance metric between the solution and the preference vector along the given direction. Although it is correct, the authors can explain this equation a bit more.
>
> **Response**.  Thanks for your suggestions. In the final version of this paper, we will formally show the derivation of Eq. (14). Specifically, we denote two points on the line by $l^i = (l_1^i, l_2^i, \ldots, l_J^i) $, and $r^i = (r_1^i, r_2^i, \ldots, r_J^i) $, respectively. The vector  $\overrightarrow{r^i l^i} = l^i - r^i = (l_1^i - r_1^i, l_2^i - r_2^i, \ldots, l_J^i - r_J^i) $. Project the vector $ \overrightarrow{r^i l^i} $ onto the direction vector $ \mathbf{v} $, and the projection coefficient is $t$, where:
> $t = \frac{\overrightarrow{l^i r^i} \cdot \mathbf{v}}{\mathbf{v} \cdot \mathbf{v}} = \frac{\sum_{j=1}^{J} (r_j^i - l_j^i) \cdot 1}{\sum_{j=1}^{J} 1^2} = \frac{\sum_{j=1}^{J} (r_j^i - l_j^i)}{\sum_{j=1}^{J} 1}$. Eq. (14) is the distance from $r^i $ to the line, which can be computed by the magnitude of the vector $\overrightarrow{r^i l^i} - t\mathbf{v} $.
>
> **Comments 2**. Specifically, the authors can give more details about Algorithm 1, such as the number of parents involved in the tournament selection, the mutation rate, and how the mutation steps are handled if the offspring exceed the predefined parameter range.
>
> **Response**. Algorithm 1 uses Monte Carlo simulation to generate $m$ points located on hyperplane $\mathcal{H}$ in each round, and then searches for the nearest Voronoi site $p_i$ for these $m$ points. The genetic algorithm is applied to optimize the objective function in Eq. (12). In genetic algorithm, we randomly select three individuals and then choose the optimal individual from them. The intersection rate $\alpha $ is uniformly distributed within the range of (0,1). The mutation method is to randomly perturb a certain point, and the range of the mutation is controlled by the parameter mutation\_std. In this article, mutation\_std is 0.05. When a newly generated point exceeds the valid range [0,1] in any dimension, the algorithm calculates a scaling factor to project the point onto the nearest boundary while preserving its original direction, provided that the directional variation component in that dimension is non-zero. This ensures that the mutated points are within a reasonable range of values.

---

> > ### Comment · Reviewer_J3SQ · 2025-04-07
> >
> > I have read the rebuttals from the authors, which have addressed most of my previous concerns.
> > Based on considering the comments from other reviewers,  I decide to raise my score.

---

> > > ### Author Response · Authors · 2025-04-08
> > >
> > > We sincerely appreciated your recognition of the work of this paper. Thank you very much!
> > >
> > > Authors of Submission16040

---

### Official Review · Reviewer_xFzh · 2025-03-09

**Overall Recommendation:** 3

**Summary:**

The paper proposes a novel sampling approach for pareto front learning and federated learning. The main idea is to use genetic algorithms to sample in a way that covers the space better. They then experiment on several MOO and FL benchmarks.

**Claims And Evidence:**

There are some issues with the evidence in this paper. Mainly, they do not report any standard deviations and when the difference is quiet small, e.g. table 7,8 and some columns in table 1, it is not clear how significant is the benefit of the method proposed here.

**Essential References Not Discussed:**

NA

**Experimental Designs Or Analyses:**

The FL section is very unclear so it is hard to evaluate the soundness of the expriments.

**Methods And Evaluation Criteria:**

Yes

**Other Comments Or Suggestions:**

Small comments:
- Explanation in beginning of Sec. 3 needs rewriting. Eq. 1 is standard optimization not PFL and it would be better to state eq. 2 in a way that would make the use of the HN clearer.
- Eq. 8 is not a good partition, this is a definition of what a partition is.
- you use "simulation points" but don't explain what they mean

**Other Strengths And Weaknesses:**

The paper offers some interesting ideas but has a few significant flaws.

The main one is the writing, specifically how the main ideas in the paper are presented.
I am still not 100% sure how alg. 1 works - once you have your optimal partition p, how do you sample point set s from it?
Also the FL part was described as a major part of the work, even part of the title, but was only described in the experimental section and was very unclear. This part should be rewritten as it is not clear at all what you are doing.

Another issue is baselines for comparison. The main point of this paper is using PFL with a modified sampling technique. However, you only compare it to the standard random sampling. To show the merit of the Voronoi sampling, it would be informative to compare to other, naive, sampling techniques. For example, sampling a large number of rays and using k-means clustering to get a small representative set.

**Questions For Authors:**

Mainly how the FL is used, do you just take an existing FL algorithm and replace the sampling?

**Relation To Broader Scientific Literature:**

It is an incremental work on preto front learning

**Theoretical Claims:**

NA

---

> ### Author Rebuttal · Authors · 2025-04-01
>
> **Notes**: All the tables can be found at https://anonymous.4open.science/r/icml_rebuttal-E75A/rebuttal.pdf.
>
> **Response to Claims and Evidence**:
>
> In the context of this paper, a special metric, namely Maximum Spread (MS),  can play a role similar to the standard deviation to better evaluate the solution's robustness [A1] below. We also carried out experiments to show the effectiveness of PHN-HVVS in terms of MS.
>
> Specifically, as done in (Hoang et al., 2023), the Hyper-Volume (HV) metric can simultaneously evaluate the convergence and diversity of a set of solutions, which refer to how closely the solutions approximate the true Pareto front (PF), and how well the solutions are spread across the entire PF, respectively. For the MS metric, a larger value indicates a wider coverage of the entire PF, reflecting superior diversity in the solution set.  Please refer to [A1] for the explanation of MS.
>
> Table 2 presents the values of MS across 6 problems. Overall, PHN-HVVS has the best performance and can better cover the entire PF. Table 3 presents the values of MS across toy examples. On convex PFs, our method achieves the largest MS.
>
> [A1] Comparison of multiobjective evolutionary algorithms: Empirical results,  EVCO'00
>
> **Response to 'how do you sample point set s from it?'**:
>
> Algorithm 1 applies a genetic algorithm to optimize the objective function in Eq. (12). The final partition approach yields n Voronoi grids and stores the Voronoi site P and simulation point set S in each grid. At this point, each s has a label for which partition it belongs to. Algorithm 2 directly utilizes the Voronoi grid structure generated by Algorithm 1, and then performs fast sampling in each round of these grids (simulated point sets with the same label) without the need for recalculation.
>
> **Response to 'baselines for comparison'**:
>
> We conducted extended experiments on the Jura and SARCOS datasets, comparing our proposed method against multiple naive sampling techniques, including: Uniform **random** sampling, **Latin** hypercube sampling, **Polar** coordinate sampling, **Dir**ichlet distribution sampling, **K-means** clustering-based representative selection. As shown in Table 4, the proposed approach outperforms other strategies.
>
> **Response to FL Part**:
>
> We appreciated your suggestions, which help enhance the paper quality. Roughly, some parameters are assumed to be known in FL, and are estimated by Pareto front learning (PFL) schemes. Optimization process is further taken in these existing FL algorithms for different purposes. The preciseness of these parameters directly affects the performance of these FL algorithms  where the PFL scheme used previously is the one  (i.e., PHN-LS) in (Navon et al., 2020). However, it can also be the scheme (i.e., PHN-HVVS) proposed in this paper. With PHN-HVVS, all these previous FL algorithms achieve a better performance since the PF is better covered.
>
> Specifically, in FL, the $n$ clients correspond to $n$ learning tasks.  The heterogeneity of data across clients entails evaluating the complementarity of data between clients. While PFL schemes are applied, there exists an optimal preference vector $p_{i}^{\ast}=${ $p_{i,1}^{\ast}, \cdots, p_{i,n}^{\ast}$} such that the model performance of $i$ can be maximized. $p_{i,j}^{\ast}$ can be used to evaluate the weight of the client $j$'s data to the model performance of client $i$. These weights define a benefit graph $G_{b}$, where there is a direct edge from $j$ to $i$ if and only if $p_{i,j}^{\ast}>0$. Several works assume that the values of $p_{1}^{\ast}, \cdots, p_{n}^{\ast}$ are known, and study how to determine a subgraph $G_{u}$ of $G_{b}$ that satisfies some desired properties to form stable coalitions, avoid conflicts of interests, or eliminate free riders. In $G_{u}$, there is a direct edge from $j$ to $i$ if $j$ will make a contribution to $i$ in the actual FL training process, and it defines the collaborative FL network.
>
> Examples of questions studied in the previous FL works are introduced in Table 5. Definitions of $G_{b}$ and  $G_{u}$ can be found in (Tan et al., 2024; Chen et al., 2024). The way of generating $p_{i}^{\ast}$ is given in Section 5.3 of this paper. In the final version , we will better clarify the FL part, including Section 2.
>
> **Response to small comments**:
>
> We will rewrite the content in the beginning of Section 3 as you suggested. Eq. (1) presents the standard optimization process. We will better state Eq. (2). The hypernetwork is explicitly described in Section 4 of (Navon et al., 2020); we will add a rephrasing  explanation in the appendix. Simulation points are introduced in Algorithm 1 of the paper.
>
> A good ideal partition should be equivalent to Eq. (8) and f=1 in the Eq. (12) where the number of points fall in each grid is equal [A4].
>
> [A4] De Berg M. Computational geometry: algorithms and applications
>
> We will update the paper according to your suggestions above.

---

> > ### Comment · Reviewer_xFzh · 2025-04-02
> >
> > I thank the authors for adding the baseline sampling methods and clarifications. I will raise my score, but there is still an important issue of missing STD to know the significance of the results not allowing me to raise it further

---

> > > ### Author Response · Authors · 2025-04-05
> > >
> > > We thank you sincerely for your recognition of the work of this paper. Regarding the experiments in Sections 5.1 and 5.2, we have conducted the training experiment for each method five times and the standard deviations of the results are provided in Table 6, which can still be found at the following link: https://anonymous.4open.science/r/icml_rebuttal-E75A/rebuttal.pdf
> > >
> > > Now,  the standard deviations of all experimental results are available. Previously, we followed the practice in the work (Hoang et al., 2023) to conduct the experiments. Your comments helped to further enhance the paper quality. We sincerely appreciated your input.

---

### Official Review · Reviewer_mv1y · 2025-03-15

**Overall Recommendation:** 1

**Summary:**

This paper proposes a method for sampling reference points (rays) from the unit simplex based on Voronoi tessellation and a genetic algorithm. Furthermore, the addition of the Hypervolume (HV) indicator to the objective function of the PHN further improves performance. The algorithm's capabilities are evaluated on synthetic functions and Federated Learning.

**Claims And Evidence:**

Please refer to "weaknesses".

**Essential References Not Discussed:**

Please refer to "weaknesses".

**Experimental Designs Or Analyses:**

Please refer to "weaknesses".

**Methods And Evaluation Criteria:**

Please refer to "weaknesses".

**Other Comments Or Suggestions:**

Please refer to "weaknesses".

**Other Strengths And Weaknesses:**

Strengths

1. This paper is well-written and easy to follow.

Weaknesses

1. The paper claims to address the challenge of sampling rays in high-dimensional space for multi-objective optimization. However, this challenge has been tackled by existing methods, such as the energy minimization approach proposed by [1], which can generate any number of rays in arbitrary dimensionality.
2. According to Algorithm 2, the rays seem to be re-sampled in each iteration. According to Algorithm 1, the generated rays are solely dependent on the dimensionality (J) and the desired ray size. Consequently, these rays could be pre-computed and reused across iterations, or even across different optimization tasks.
3. The empirical study primarily focuses on 2-3 objective optimization problems. Since the proposed sampling method is particularly applicable to high-dimensional situations, it is suggested to include more high-dimensional problems.
4. The combination of the proposed ray sampling method with the  HV does not seem reasonable. HV can not be accurately calculated in high-dimensional spaces.  While the proposed ray sampling method aims to address challenges associated with high dimensionality, the reliance on HV as a performance metric negates its potential benefits. Therefore, it appears that at least one of the contributions, either the ray sampling method or HV is useless.
5. The authors claim a challenge exists in covering convex Pareto Fronts. However, this issue has been well-addressed for decades [2], and this paper does not propose new methods for this challenge.
6. In Figure 5, the points are overlapping and difficult to distinguish.

[1] Generating well-spaced points on a unit simplex for evolutionary many-objective optimization, IEEE TEVC, 2020.

[2] MOEA/D: A multiobjective evolutionary algorithm based on decomposition, IEEE TEVC, 2007.

**Questions For Authors:**

Please refer to "weaknesses".

**Relation To Broader Scientific Literature:**

N/A.

**Theoretical Claims:**

This paper does not include theoretical claims.

---

> ### Author Rebuttal · Authors · 2025-04-01
>
> **Notes**: All the tables and figures can be found at https://anonymous.4open.science/r/icml_rebuttal-E75A/rebuttal.pdf.
>
> **Response to W1 \& W2**:  We appreciated your insightful questions, which ever motivated us to develop the solution of this paper.
>
> Specifically, there is a mapping of rays to solutions. We aim to find a set of Pareto-optimal solutions that can well cover the entire Pareto front (PF). We fully agree on that the Riesz s-energy based iterative optimization method in [1] can generate uniformly distributed points or rays in high-dimensional spaces. However, a set of uniformly distributed rays does not necessarily lead to a set of uniformly distributed solutions, as shown in [A1] below and Figure 1. When rays are generated in advance, using such fixed rays generated by [1] may fail to cover some parts of the PF.
>
> Thus, we propose to dynamically sample the preference vector during the optimization process, so that the hypernetwork technology can continuously explore the uncovered areas of the target space. The method proposed in this paper does not directly generate a globally uniform point set, but instead performs preference vector re-sampling based on local units of Voronoi grids generated by Alg. 1. Even with a fixed set of points generated by [1], generating preference vectors within the cell formed by these points still faces challenges, such as the geometric complexity problem in high-dimensional space [A2].
>
> Above, we explain why dynamic sampling of rays is used. In the ray sampling process, optimization is also taken in Alg. 1 and 2 to avoid recalculation, like your suggestion. Specifically, Alg. 1 uses Monte Carlo simulation to generate m simulation points S located on hyperplane $\mathcal{H}$ at each iteration:  $\mathcal{H} =${$(x_1, \ldots, x_J) \in \mathbb{R}^J \mid x_1 + \ldots + x_J = 1$ } and then searches for the nearest Voronoi site  $p_i \in P =$ {$p_1, p_2, \ldots, p_n $} for these m points. The genetic algorithm is used to find the partition method with the maximum objective function, ultimately obtaining n Voronoi grids (while storing the simulation point sets of each grid). Alg. 2 directly utilizes the Voronoi grid structure generated by Alg. 1 and performs fast sampling directly in each round of these grids (simulation point sets) without the need for recalculation.
>
>
> [A1] Multi-objective deep learning with adaptive reference vectors, NeurIPS'22
>
> [A2] An optimal convex hull algorithm in any fixed dimension, DCG'93
>
> **Response to W3**：We have added experiments on the DTLZ1 benchmark problem [A3]: we measured the results for 4, 5, and 6 objectives, respectively, where the reference points were set as (2,..., 2). It can be seen from Table 1 that our method achieves the best result.
>
> [A3] Deb et al. Scalable multi-objective optimization test problems
>
> **Response to W4**：In high-dimensional space, HV can be effectively approximated [A4], and there is a standard indicators .hv in Python's Pymoo library used extensively for compute HV. For the error of HV, when J>3, the module in the library uses Monte Carlo method to sample about 10000 samples to estimate the HV value. The error rate decreases with the square root of the sample size (i.e. $\frac{1}{\sqrt{N}}$), and the actual error can be controlled within the range of 1\% to 5\%.
>
> In this paper, the value of this HV is only computed once and used for the final evaluation of algorithm performance. We do not calculate the specific value of HV every round, but instead use gradient descent method to obtain the $\phi$ that minimize Eq. (13). The HV gradient computation of this paper follows (Hoang et al., 2023,Wang et al., 2017, Emmerich & Deutz, 2014).
>
> [A4] HypE: An algorithm for fast hypervolume-based many-objective optimization, EVCO'11
>
> **Response to W5**: To address your concern, we will better clarify in the final version that this paper focuses on the emerging paradigm of Pareto Front Learning (PFL). Compared with traditional MOEAs, PFL has its own challenges to be addressed specially.
>
> Specifically, traditional MOEAs rely on the diversity of evolutionary population to search PF. Variants of MOEA/D and NSGA-III can address convex problems by adopting adaptive reference vectors. Differently, PFL typically uses a hypernetwork (HN) to approximate the PF. Existing PFL methods employ gradient optimization to maximize HV by approximating gradients with HV contributions. However, as shown in (Zhang et al., 2023), convex PF boundary solutions suffer weight decay: intermediate solutions have larger HV gradients, causing preferential fitting of central regions. In traditional MOO, such weight decay does not need to be addressed, and MOEAs directly search boundary solutions through population diversity maintenance, without gradient reliance in PFL.
>
>
> **Response to W6**: We have optimized the visualization; please see Figure 2 in the link.
>
> In the final version of this paper, we will better clarify the above contents.

---

> > ### Comment · Reviewer_mv1y · 2025-04-03
> >
> > Thank you for your detailed response. However, I still have some concerns, especially about the motivation of Voronoi sampling.
> >
> > R1 & R2: I am still very confused about the motivation of Voronoi sampling, which seems to be the core contribution of this paper. I agree that a set of uniformly distributed rays does not necessarily lead to a set of uniformly distributed solutions, and some techniques like weight adaptation may solve this problem. However, from Lines 171-173 and Lines 199-201, it seems that the only thing Algorithm 1 does is sample a set of uniformly distributed rays in a unit simplex $\mathcal{H}$. The input of Algorithm 1 is the dimensionality and size. So, if we don't take randomness into account, for the same problem, the output of Algorithm 1 will be identical. From your response, it seems that you are using dynamic sampling to introduce some randomness to help the exploration of some part of the PF, so why not use pure random sampling? In conclusion, I believe that the motivation and function of Voronoi sampling have not been clearly explained, and there lack of empirical results (such as ablations) to support the significance of Voronoi sampling.
> >
> > R3: These results are impressive. However, the PF of DTLZ1 is a unit simplex, so it does not reveal the advantage of your proposed "dynamically sample the preference vector". Why not try the complete DTLZ suite, i.e., DTLZ1-7? I think running on such synthetic problems is not very costly.
> >
> > R4: I agree that HV is only calculated once, and the gradient of HV is calculated in each iteration. However, to my knowledge, calculating the gradient of HV is still difficult in high dimensionality and is not much faster than calculating HV itself [1]. I agree that Mont Carlo is always a workaround but if I remember correctly, exact calculation in high dimensionality is not feasible now.
> >
> > R5: I think the phenomenon you mentioned is not the inherent challenge of PFL, but a limitation of HV. It also applies to traditional MOAs that adopt HV maximization [2]. Moreover, how HV addresses convexity is also related to the selection of the reference point. Given that your method also relies on HV maximization, this challenge does not seem to be addressed, and no empirical evidence in this paper demonstrates the performance on convex PFs. I suggest trying the 3-objective DTLZ2 and presenting the visualized result of the PF approximation.
> >
> > R6: The improved figures seem much better. Good job.
> >
> > References
> >
> > [1] Emmerich, Michael, and André Deutz. "Time complexity and zeros of the hypervolume indicator gradient field."
> >
> > [2] A Survey on the Hypervolume Indicator in Evolutionary Multiobjective Optimization. IEEE TEVC 2020.

---

> > > ### Author Response · Authors · 2025-04-08
> > >
> > > We sincerely thank you for your time in reviewing our paper. As the deadline is approaching, we would greatly appreciate it if you could let us know whether the responses provided below have addressed your concerns.
> > >
> > > All the tables and figures below can be found at the link: https://anonymous.4open.science/r/icml_rebuttal-E75A/rebuttal.pdf.
> > >
> > > **Response to R1 & R2**：Algorithm 1 is called in line 5 of Algorithm 2. It outputs a fixed Voronoi partition of the hyperplane $\mathcal{H}$, which generates spatial partitions based on  Voronoi sites $P=${$p_{1}, p_{2}, \cdots, p_{n}$} (while also storing simulation points $S=${$s_{1}, s_{2}, \cdots, s_{m}$} within different partitions), rather than generating fixed preference vectors or rays $r$. Each cell/grid $V(p_{i})$ is equivalent to a subregion $\Omega_{i}$. In Algorithm 2, we randomly sample one ray from each cell/grid in the last line of Algorithm 1. As explained in (Hoang et al., 2023), adopting partition can make the HV and penalty functions work better. Specifically, partitioning improves the effectiveness of HV and penalty functions. In 2D scenarios, Hoang et al. (2023) uses uniform partitioning, but high-dimensional cases face challenges: the number $p$ of rays for HV hypernetworks cannot be freely set, and the partition count grows exponentially with J and k, causing severe computational complexity. To address this, the Dirichlet distribution is typically adopted in [A1] and (Navon et al., 2020; Hoang et al., 2023). Our work employs a Voronoi grid distribution for high dimensions, storing simulation points within each grid and performing round-by-round sampling as outlined in Algorithm 2. Each round of dynamic sampling is to explore the entire space and obtain a complete PF. Voronoi partition ensures the coverage of each region during the sampling process by dividing the hyperplane $\mathcal{H}$ into multiple sub regions $\Omega_{i}$ (each consisting of the nearest neighbors of the corresponding sites). Specifically, the sample points within each Voronoi region are closest to the site of that region, which naturally avoids the problem of local aggregation or omission in the sampling distribution. This partition method ensures effective exploration of the global PF. Random sampling cannot achieve sampling from the entire space. The method in [1] generates uniform sampling, but because the preference vector is completely fixed and there is no randomness, it is difficult to obtain uniformly distributed Pareto optimal solutions when dealing with irregular frontiers such as convex shapes. In Table 4, we compare different sampling methods, including random sampling, and it can be seen that Voronoi partitioning sampling achieves the best results.
> > >
> > > The effectiveness of our approach is also validated by extensive experiments, including not only the HV values but also the Pareto fornts and its improvement to the performance of three FL frameworks.
> > >
> > >
> > > [A1] Tuan et al. A framework for controllable pareto front learning with completed scalarization functions and its applications.
> > >
> > > **Response to R3**：We have conducted experiments using the complete DTLz suite, which helped to validate the advantage of our proposed "dynamically sample the preference vector". The experimental results are presented in Tables 7, 8, and 9. Your comments helped to enhance our experimental design and the paper quality, thanks.
> > >
> > > **Response to R4**：Yes, accurately calculating HV in high dimensions is not feasible. However, it can be effectively approximated. We use Python's built-in pymoo HV class, which is used extensively in practice and can control the actual error within 1\% to 5\% [A2,A3]. We agree that it is difficult to calculate the gradient of HV in high dimensions, so we use HV contribution approach to approximate HV gradient (Wang et al., 2017), as done in [A4,A5] and (Hoang et al., 2023). Specifically, because the algorithm framework incorporates hypernetwork technology, we do not focus on accurately calculating the specific value of HV, but on calculating the gradient of HV every round to obtain parameters $\phi$. Similarly, We do not calculate the accurate value of HV gradient, but use HV contribution to approximate it. The effectiveness of this approach in high-dimensional space is also verified in the experiments of (Hoang et al., 2023).
> > >
> > > [A2] Bader et al. HypE: An algorithm for fast hypervolume-based many-objective optimization.
> > >
> > > [A3] Blank et al. Pymoo: Multi-objective optimization in python.
> > >
> > > [A4] Deist et al. Multi-objective learning using hv maximization.
> > >
> > > [A5] Liu et al. Profiling pareto front with multi-objective stein variational gradient descent.
> > >
> > > **Response to R5**：Following your suggestions, we conducted additional experiments using the 3-objective DTLZ2 benchmark, which is known for its convex Pareto front (PF). The results are now presented in Figure 3. These empirical evidences helped to demonstrate the performance on convex PFs.

---

### Decision · Program_Chairs · 2025-05-01

**Decision:**

Accept (poster)

**Comment:**

Strength:

1. The idea of applying the Pareto front (PF) to the FL is interesting and novel.

2. The paper is theoretically sound, which introduces the Voronoi paritition to help find solutions.

Weakness

1. Although the application is novel, the Pareto Front technique is directly applied with incremental modifications, which is limited in its novelty.

2.  The motivation of the Voronoi paritition is not clear, which appears pursuing greater uniformity.

3. The experimental improvement is small.